# Generalized Tangent Kernel: A Unified Geometric Foundation for Natural Gradient and Standard Gradient

**Qinxun Bai**[*]
*Horizon Robotics, Cupertino, CA, USA*

*qinxun.bai@horizon.auto*

**Steven Rosenberg**[*]
*Department of Mathematics and Statistics, Boston University, Boston, MA, USA*

*sr@math.bu.edu*

**Wei Xu**
*Horizon Robotics, Cupertino, CA, USA*

*wei.xu@horizon.auto*

**Reviewed on OpenReview:** *https://openreview.net/forum?id=HOnL5hjaIt*

## Abstract

Natural gradients have been widely studied from both theoretical and empirical perspectives, and it is commonly believed that natural gradients have advantages over standard (Euclidean) gradients in capturing the intrinsic geometric structure of the underlying function space and being invariant under reparameterization. However, for function optimization, a fundamental theoretical issue regarding the existence of natural gradients on the function space remains underexplored. We address this issue by providing a geometric perspective and mathematical framework for studying both natural gradient and standard gradient that is more complete than existing studies. The key tool that unifies natural gradient and standard gradient is a generalized form of the Neural Tangent Kernel (NTK), which we name the Generalized Tangent Kernel (GTK). Using a novel orthonormality property of GTK, we show that for a fixed parameterization, GTK determines a Riemannian metric on the entire function space which makes the standard gradient as "natural" as the natural gradient in capturing the intrinsic structure of the parameterized function space. Many aspects of this approach relate to RKHS theory. For the practical side of this theory paper, we showcase that our framework motivates new solutions to the non-immersion/degenerate case of natural gradient and leads to new families of natural/standard gradient descent methods.

## 1 Introduction

Since their introduction in Amari (1998), natural gradients have been a topic of interest among both theoretical researchers (Ollivier, 2015; Martens, 2014; Li & Montúfar, 2018) and practitioners (Kakade, 2001; Pascanu & Bengio, 2013). In recent years, research has examined several important theoretical aspects of natural gradient for function optimization on neural networks, such as replacing the KL-divergence by the $L^2$ metric on the function space (Benjamin et al., 2019), Riemannian metrics for neural networks (Ollivier, 2015), and convergence properties under overparameterization (Zhang et al., 2019b).

It is well-known that natural gradient has the favorable property of being invariant under change of coordinates. While this invariance property is appealing theoretically, in modern practice of (stochastic) gradient descent, the coordinates are rarely changed during the gradient descent procedure. Therefore, in this work, we focus on the properties of natural gradient and standard gradient under some fixed function parameterization and coordinates. In this setup, it is widely believed that natural gradients have an advantage over standard (Euclidean) [1] gradients in capturing the intrinsic geometric structure of the parameterized function space.

---

[*]Correspondence. These authors contributed equally to this work.
[1]Throughout the paper, we use *standard gradient* to denote the gradient for the standard Euclidean metric on the parameter space, which is the default choice for (stochastic) gradient descent in most machine learning practice.

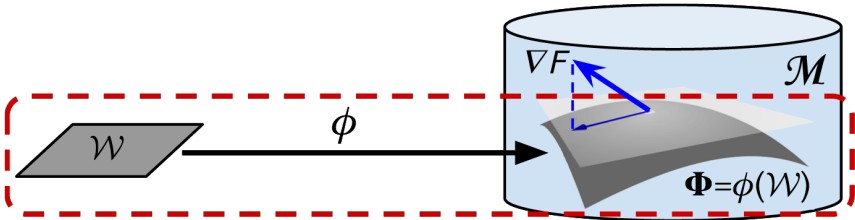

Figure 1: Most existing work on natural gradient focuses on the red dashed regime, where $\phi$ is a mapping from the parameter space $\mathcal{W}$ to the parameterized function space $\Phi = \phi(\mathcal{W})$. Natural gradient decreases the objective $F$ along the gradient flow on $\Phi$ under a chosen Riemannian metric, instead of using the standard Euclidean metric on $\mathcal{W}$. However, the functional gradient $\nabla F$ at any point of $\Phi$ need not lie in the tangent space to $\Phi$.

However, some critical theoretical aspects of natural gradients related to function approximations are still largely underexplored. In parametric approaches to machine learning, one often maps the parameter domain $\mathcal{W} \subset \mathbb{R}^k$ to a space $\mathcal{M} = \mathcal{M}(M, N)$ of differentiable functions from a manifold $M$ to another manifold $N$. For example, in neural networks, $\mathcal{W}$ is the space of network weights, and $\phi \colon \mathcal{W} \to \mathcal{M}(\mathbb{R}^n, \mathbb{R}^m)$ has $\phi(w)$ equal to the associated network function, where $n$ is the dimension of network inputs and $m$ the dimension of outputs. In the modern formulation of natural gradient, if $\phi$ is an immersion, a Riemannian metric on $\mathcal{M}$ induces a pullback or natural metric on $\mathcal{W}$. In supervised training of such neural networks, the empirical loss $F$ of a particular type, say mean squared error (MSE) for regression and cross-entropy for classification, is typically applied to the parametric function evaluated on a finite set of training data. As shown in Figure 1, most existing work (with a notable exception of van Oostrum et al. (2023)) on natural gradient focuses on the red dashed regime, where natural gradient decreases the objective $F$ along the gradient flow on $\Phi = \phi(\mathcal{W})$ under the natural metric, instead of using the standard Euclidean metric on $\mathcal{W}$. However, the functional gradient[2] $\nabla F$ at any point of $\Phi$ need not lie in the tangent space to $\Phi$. Therefore, a complete study of the natural gradient should analyze both the gradient of $F$ in the ambient function space $\mathcal{M} \supset \Phi$ and its projection onto $\Phi$, before pulling it back to $\mathcal{W}$. This motivates our study of the analytic and differential geometric properties of $\mathcal{M}$ that influence the projection and the pullback metric. Moreover, we can interpret this figure to encompass a family of projected functional gradients, including natural gradient and standard gradient, which leads to new understandings of both.

Based on this geometric viewpoint, we address the relationship between the gradient flow on $\Phi$ and that on $\mathcal{M}$, showing that the inverse of the metric matrix appearing in the natural gradient is in fact a consequence of the orthogonal projection of the functional gradient from $\mathcal{M}$ to $\Phi$. We also show that because of the "pointwise" nature of a typical empirical loss, the gradient of the loss function does not in general exist on the infinite-dimensional space $\mathcal{M}$ for common choices of metrics, such as $L^2$ and Fisher-Rao. The technical issue is that any such gradient involves delta functions, which are not continuous linear functionals on *e.g.,* $L^2$. This motivates studying the pullback metric in an RKHS, where by definition delta functions are continuous linear functionals. In particular, for all common empirical loss functions in supervised learning, the gradient is well-defined only in the RKHS context. Using an RKHS framework, we introduce a generalized form of the Neural Tangent Kernel (NTK) (Jacot et al., 2019), which we name the Generalized Tangent Kernel (GTK), to mathematically express Figure 1's common framework of natural gradient and standard gradient.

In particular, we show that GTK enjoys an automatic orthonormality property, and as a result, there exists a family of projected functional gradients (similar to the projection of $F$ in Figure 1), each of which produces a natural gradient that reflects a well-defined geometric structure on the immersed function space. Standard gradient is the simplest example of this family in the computational sense that the metric matrix is the identity matrix.

We then study a known issue of applying natural gradients to modern function approximations, such as neural networks, namely the potential violation of the immersion/injectivity assumption on $\phi$, which leads to a

---

[2]By "functional gradient," we mean the gradient of a loss function on a space of functions.

singular metric matrix and therefore an ill-defined natural gradient on $\mathcal{W}$. In contrast to existing approaches resorting to the Moore-Penrose generalized inverse (Ollivier, 2015; Bernacchia et al., 2018; van Oostrum et al., 2023), we use slice methods to reduce the dimension of $\mathcal{W}$ and show that our GTK-based framework provides a new solution to this issue, leading to new variants of natural/standard gradient for exemplar neural networks.

To summarize the main theoretical results of the paper, whenever we have a pointwise defined loss function on a Riemannian Hilbert space/manifold $\mathcal{M}$ of functions, the functional gradient exists on $\mathcal{M}$ iff each $T_f\mathcal{M}, f \in \mathcal{M}$, is an RKHS. For a finite dimensional parameterized function space $\Phi \subset \mathcal{M}$, the existence of a natural gradient on $\Phi$ (and thus on the parameter space) is then determined by a projection from $T_f\mathcal{M}$ to $T_f\Phi$. The existence of such a projection is unexpectedly universal and flexible, and is encoded by GTK theory, which produces a family of projected RKHS structures on each $T_f\Phi, f \in \Phi$, and unifies classic natural gradient, standard gradient, and beyond. We believe this is a more complete understanding of the geometric foundation of projected functional gradients than has appeared in the machine learning literature. Viewing both natural and standard gradients through the lens of GTK also sheds light on why standard gradient (and thus SGD) work so well in modern machine learning practice, while the advantage of natural gradient is often minimal.[3].

As an outline of the paper, in §2 we introduce the theoretical setup, schematically pictured in Figure 1, which establishes a projection viewpoint for both natural gradient and standard gradient. In §3, we provide a careful analysis of the non-existence of natural gradients for empirical loss functions on commonly used function spaces. This theoretical issue is addressed in §4 by working in an RKHS, which motivates our core concept, the Generalized Tangent Kernel (GTK). The automatic orthonormality property of GTK (Theorem 4.2) produces a family of projections of the functional gradient on $\mathcal{M}$ onto the tangent space $\Phi$ via RKHS theory, with natural gradient and standard gradient as special cases. A new variant of natural gradient is also motivated by this RKHS framework; see Appendix G. In §5, we study the ill-defined natural gradients for non-immersion function approximations widely used in machine learning practice, such as neural networks with (piecewise) linear activations. We use a GTK-based framework to study the slice/reduction of $\phi$ for ReLU MLP as an example and derive new variants of standard gradient algorithms. Preliminary experimental results of these new algorithms are also reported. We summarize this work and discuss future directions in §6. Background material, proofs, and preliminary experiments for exemplar new natural gradient variants motivated by the GTK framework are in the Appendices.

Because of the technical nature of the paper, we encourage the reader to refer to Table 1 in Appendix A for key mathematical notation, and to Appendix B for background materials on Riemannian manifolds, Sobolev spaces, and RKHS theory.

## 2 The Theoretical Setup: A Geometric Perspective

In supervised learning, we typically want to learn an optimal function $f\colon \mathbb{R}^n \to \mathbb{R}^m$, or more generally $f\colon M \to N$, where $M, N$ are $n$-dimensional, $m$-dimensional manifolds, respectively. Here an optimal function minimizes some loss function $F\colon \mathcal{M} \to \mathbb{R}$, where $\mathcal{M}$ is the space of $f$'s.

When function approximation is applied, $f$ is parameterized by some parameter vector of finite dimension, and the general situation can be described by the following simple diagram:

$$\begin{array}{ccc} \mathcal{W} & \xrightarrow{\ \phi\ } & \mathcal{M} \\ & \underset{\widetilde{F}}{\searrow} & \downarrow{F} \\ & & \mathbb{R} \end{array} \tag{1}$$

Here $\mathcal{M} = \mathcal{M}(M, N)$ is a space of smooth maps from $M$ to $N$. $\mathcal{M}$ is an infinite dimensional manifold with a reasonable (high Sobolev or Fréchet) topology. The parameter space $\mathcal{W}$, also a manifold, is usually a compact

---

[3]For instance, Zhang et al. (2019a) empirically observes that the advantages of natural gradient over standard gradient are minimal except for impractically large training batchsize, even without worrying about the computational cost.

subset of a Euclidean space, $\phi\colon \mathcal{W} \to \mathcal{M}$ is the parametrization of the image $\phi(\mathcal{W}) = \Phi \subset \mathcal{M}$, $F$ is the loss function, and $\widetilde{F} = F \circ \phi$ makes the diagram commute.

In this section, we always assume that $\phi$ is an immersion, *i.e.,* at every $w \in \mathcal{W}$, the differential $d\phi_w\colon T_w\mathcal{W} \to T_{\phi(w)}\Phi$ is injective, where $T_w\mathcal{W}$ is the tangent space to $\mathcal{W}$ at $w$, and similarly for $T_{\phi(w)}\Phi$. Then for a fixed Riemannian metric $\bar{g}$ on $\mathcal{M}$, the *pullback metric* on $X, Y \in T_w\mathcal{W}$ is given by

$$(\phi^*\bar{g})(X, Y) := \bar{g}(d\phi(X), d\phi(Y)). \tag{2}$$

The case where $\phi$ is not an immersion is discussed in §5, giving an alternative to using the Moore-Penrose inverse as in van Oostrum et al. (2023). Proofs for this section are in Appendix C.

### 2.1 Smooth Gradients, Orthogonal Projection, and Pullback Metric for Natural Gradient

For a metric $\bar{g}$ on $\mathcal{M}$, we want to study the gradient flow of $F$ on the image $\Phi$, and this should be equivalent to studying gradient flow for $\widetilde{F}$ on $\mathcal{W}$ via the pullback metric (2). The existence of $\nabla F$ on $\mathcal{M}$ is carefully studied in §3. For now, assuming that $\nabla F$ exists on $\mathcal{M}$, we still have to distinguish between the gradient flow lines of $F$ in $\mathcal{M}$ and the flow lines on $\Phi$. Indeed, a gradient flow line of $F$ starting in $\Phi$ will not stay in $\Phi$ in general, since $\nabla F$ need not point in $\Phi$ directions.

The following lemma establishes the relationship between gradient flow in $\mathcal{M}$ and flows on $\Phi$, and the equivalence between pullback gradients on the parameter space $\mathcal{W}$ and gradients on $\Phi$ in the function space.

**Lemma 2.1.** *(i) The gradient flow on $\Phi$ for $F|_\Phi$ is given by the flow of $P(\nabla F)$, where $P = P_{\phi(w)}$ is the orthogonal projection from $T_{\phi(w)}\mathcal{M}$ to $T_{\phi(w)}\Phi$ with respect to a Riemannian metric $\bar{g}$ on $\mathcal{M}$.*

*(ii) Let $\widetilde{g} = \phi^*\bar{g}$ be the pullback metric on $\mathcal{W}$ given by (2). Then $d\phi(\nabla_{\widetilde{g}}\widetilde{F}) = \nabla_{\bar{g}}F$, and $\gamma(t)$ is a gradient flow line of $\widetilde{F}$ with respect to $\widetilde{g}$ iff $\phi(\gamma(t))$ is a gradient flow line of $F|_\Phi$.*

By this Lemma, to get $\nabla\widetilde{F}$ on $\mathcal{W}$, we just need to compute $P(\nabla F)$. Starting below, we use the Einstein summation convention of summing over repeated indices that occur once as a subscript and once as a superscript, so *e.g.,* $g^{ij}v_iw_j = \sum_{ij} g^{ij}v_iw_j$, $\langle \nabla F, h_i \rangle h_i = \sum_i \langle \nabla F, h_i \rangle h_i$.

**Proposition 2.1.** *The orthogonal projection of $\nabla F$ onto $\Phi$ is*

$$P(\nabla F) = \widetilde{g}^{ij} \left\langle \nabla F, \frac{\partial \phi}{\partial w^i} \right\rangle \frac{\partial \phi}{\partial w^j}, \tag{3}$$

*where $(\widetilde{g}_{ij}) = \widetilde{g} = \phi^*\bar{g}$ is the pullback metric on $\mathcal{W}$, $(\widetilde{g}^{ij})$ is its inverse matrix, $\{(w^i)\}$ are the coordinates on $\mathcal{W}$, and $\partial\phi/\partial w^i = d\phi(\partial/\partial w^i)$.*

Proposition 2.1 shows that the appearance of the inverse metric matrix $(\widetilde{g}^{ij})$ in natural gradients is in fact a consequence of the orthogonal projection of $\nabla F$ from $\mathcal{M}$ to $\Phi$. This projection view becomes clearer when compared with standard gradient where the Euclidean metric is used, which we discuss in the next section.

**Remark 2.1.** Instead of (3), we can first obtain an orthonormal basis $\{h_i\}_{i=1,\ldots\dim(\mathcal{W})}$ of $T_f\Phi$, say, by applying Gram-Schmidt to the pushforward basis $\{\frac{\partial\phi}{\partial w^i}\}$ of $T_f\Phi$, then compute the orthogonal projection of $\nabla F$ onto $\Phi$ by

$$P(\nabla F) = \sum_i \langle \nabla F, h_i \rangle h_i. \tag{4}$$

**Example 2.2.** *We give a simple example for the results in this section. Let $\phi\colon W = (0, 2\pi) \times (0, \pi/2) \to \mathbb{R}^3$, $(\theta, \phi) = (\cos\theta\sin\phi, \sin\theta\sin\phi, \cos\phi)$ be the usual spherical coordinates on $S^2$. Then $\partial/\partial\theta = (-\sin\theta\sin\phi, \cos\theta\sin\phi, 0)$, $\partial/\partial\phi = (\cos\theta\cos\phi, \sin\theta\cos\phi, -\sin\phi)$, and the induced metric $\phi^*\bar{g}$ on $W$ for the standard dot product $\bar{g}$ on $\mathbb{R}^3$ is*

$$\tilde{g} = \phi^*\bar{g} = \begin{pmatrix} \sin^2\phi & 0 \\ 0 & 1 \end{pmatrix}.$$

*Then (3) becomes*

$$P(\nabla F) = \csc^2(\phi)(\nabla F \cdot (-\sin\theta\sin\phi, \cos\theta\sin\phi, 0))\frac{\partial}{\partial\theta} + \nabla F \cdot (-\cos\theta\cos\phi, \sin\theta\cos\phi, -\sin\phi)\frac{\partial}{\partial\phi}$$

$$= \csc^2(\phi)\left(-F_x\sin\theta\sin\phi + F_y\cos\theta\sin\phi\right)\frac{\partial}{\partial\theta} + \left(-F_x\cos\theta\cos\phi + F_y\sin\theta\cos\phi - F_z\sin\phi\right)\frac{\partial}{\partial\phi}.$$

## 2.2 Surjection and Pushforward Metric for Standard Gradient

In contrast to Proposition 2.1, where the gradient on $\Phi$ is obtained by a projection, standard gradient can be obtained from a surjection $S$ from $T_f\mathcal{M}$ to $T_f\Phi$ given by

$$S(\nabla F) = \left\langle \nabla F, \frac{\partial\phi}{\partial w^i} \right\rangle \frac{\partial\phi}{\partial w^i}. \tag{5}$$

While $S$ is similar to the orthogonal projection (3) in that $S|_{(T_f\Phi)^\perp} = 0$, $S$ is not a projection: $S^2 \neq S$, and $S|_{T_f\Phi}$ is not the identity map. Thus Lemma 2.1(i) fails, and the flow lines for $\nabla(F|_\Phi)$ are not related to the flow lines of $S\nabla F$. As will be shown in §4.2, $S$ becomes an orthogonal projection iff we had a Riemannian metric of $\mathcal{M}$ in directions tangent to $\Phi$ such that:

$$\left\langle \frac{\partial\phi}{\partial w^i}, \frac{\partial\phi}{\partial w^j} \right\rangle = \delta_{ij}. \tag{6}$$

In other words, $\{\frac{\partial\phi}{\partial w^i}\}$ must be an orthonormal basis of $T_f\Phi$, which is in general not true. Nevertheless, under our projection view, standard gradient uses the metric (6) on $\Phi$, which, by the following Remark, implicitly uses the *pushforward* $\phi_* g_E$ of the Euclidean metric from $\mathcal{W}$ to $\Phi$.

**Remark 2.3.** *For a smooth map $\phi\colon \mathcal{W} \to \mathcal{M}$, the pushforward of a metric $g$ on $\mathcal{W}$ to a metric $\phi_* g$ on $\Phi$ is given by*

$$(\phi_* g)(X, Y) := g((d\phi)^{-1}X, (d\phi)^{-1}Y), \ \ X, Y \in T\Phi.$$

*$\phi_* g$ exists iff $\phi$ is an immersion. Since $d\phi(\frac{\partial}{\partial w^i}) = \frac{\partial\phi}{\partial w^i}$ and $g_E\left(\frac{\partial}{\partial w^i}, \frac{\partial}{\partial w^j}\right) = \delta_{ij}$, the Riemannian metric determined by (6) is the pushforward $\phi_* g_E$ of the Euclidean metric on $\mathcal{W}$.*

# 3 Natural Gradients of the Empirical Loss

Based on the theoretical setup of §2, in this section, we analyze the non-existence issue of natural gradients for general empirical losses on commonly used function spaces. We consider the most commonly used setting of $\mathcal{M}$ in supervised learning, where the target space $N$ is $\mathbb{R}^n$, and the loss function $F$ is some empirical loss. In this case, $\mathcal{M}$ is an infinite dimensional vector space, so we have to specify its topology. For the easiest topology, the one induced by the $L^2$ inner product, we show that the gradient of the loss function does not exist in any strict mathematical sense. More precisely, a follow-your-nose computation of the gradient produces an expression containing delta functions, which are not elements of $L^2$. To make this computation mathematically rigorous, we have to change the topology/inner product so that delta functions are *continuous* linear functionals on the function space. This property of delta functions characterize RKHSs, which motivates the use of RKHS theory in §4.

A typical choice of empirical loss term on $\mathcal{M}$ is the mean squared error (MSE)

$$\ell_E(f) = \frac{1}{2}\sum_{i=1}^{k}\|f(x_i) - y_i\|^2 \tag{7}$$

for a set $\{(x_i, y_i)\}_{i=1}^k \subset M \times \mathbb{R}^n$ of training data. Let $f\colon M \to \mathbb{R}^n$ have components $f^j\colon M \to \mathbb{R}$, and let $\delta_i$ be the "$\delta$-vector field", $\delta_i(f^j) = f^j(x_i) \in \mathbb{R}$. We formally[4] and incorrectly consider $\delta_i$ to be a function on $M$

---

[4]Throughout the paper, "formal/formally" means that a concept does not have a mathematically sound definition, but can be manipulated mechanically.

with the following $L^2$ inner product,

$$\langle \delta_i, r \rangle_{L^2} = \int_M \delta_i r = r(x_i), \tag{8}$$

for $r \in C^\infty(M)$. The integral is with respect to some volume form on $M$.

Take a curve $(f_t)_{t \geq 0} \in \mathcal{M}$ with $f_0 = f$, and set $h = \dot{f}_t|_{t=0} \in T_f \mathcal{M} \simeq \mathcal{M}$. In the following formal computation, we ignore several nontrivial technicalities, which are detailed in Appendix D. The differential $d\ell_E$ at $f$, denoted by $d\ell_{E,f} \colon T_f \mathcal{M} \to \mathbb{R}$, satisfies

$$d\ell_{E,f}(h) = \frac{d}{dt}\bigg|_{t=0} \frac{1}{2} \sum_i \|f_t(x_i) - y_i\|^2 = \sum_i \sum_{j=1}^n (f^j(x_i) - y_i^j)h^j(x_i). \tag{9}$$

Thus we formally get

$$d\ell_{E,f}(h) = \int_M \sum_j \left( \sum_i (f^j(x_i) - y_i^j)\delta_i \right) h^j. \tag{10}$$

By the definition of the gradient, for $\nabla \ell_{E,f}$ to exist under the $L^2$ inner product (8), we must have

$$d\ell_{E,f}(h) = \langle \nabla \ell_{E,f}, h \rangle_{L^2} = \int_M \sum_j (\nabla \ell_{E,f})^j \cdot_E h^j, \tag{11}$$

where $\cdot_E$ is the Euclidean dot product. Comparing (10) and (11), the formal gradient of $\ell_E$ at $f$ has $j^{\text{th}}$ component given by

$$(\nabla \ell_{E,f})^j = \sum_i (f(x_i) - y_i)^j \delta_i. \tag{12}$$

In other words, the gradient of the empirical loss (7), if it exists, must formally be a sum of delta functions. Since delta functions are not $L^2$ functions, *the $L^2$ gradient does not exist on $\mathcal{M}$.* Thus we cannot apply Prop. 2.1 to compute the gradient of $f$ on $\Phi$, even though the gradient on the finite dimensional manifold $\Phi$ must exist.

**Remark 3.1.** *We emphasize that the non-existence of the $L^2$ gradient is endemic to supervised learning, because of the discrete nature of empirical loss function. For a general differentiable function $L = L(z,w) \colon \mathbb{R}^n \times \mathbb{R}^n \to \mathbb{R}$, the corresponding $L$-empirical loss is $\ell_L(f) = \sum_i L(f(x_i), y_i)$. For $\nabla^E L_z$ the Euclidean gradient in the $z$ direction, the formal gradient (12) becomes $(\nabla \ell_{L,f})^j = \sum_i \left( \nabla^E L_z(f(x_i), y_i) \right)^j \delta_i$, which again is a sum of delta functions.*

As another common example in machine learning, we consider the Fisher-Rao metric on the space of $C^\infty$ probability distributions $\mathcal{P} = \{p \in C^\infty(M, \mathbb{R}) : p > 0, \int_M p(x)d\mu(x) = 1\}$ on a measure space $(M, \mu)$, where $M$ is a compact manifold without boundary. The tangent space at $p \in \mathcal{P}$ is $T_p\mathcal{P} = \{h \in C^\infty(M, \mathbb{R}) : \int_M h d\mu = 0\}$ (Lafferty, 1988, §3). Let $\phi \colon \mathcal{W} \to \mathcal{P}$ give a parametrized submanifold $\Phi = \phi(\mathcal{W}) \subset \mathcal{P}$, with $\phi(w)(x) := p(x, w)$. The Fisher-Rao metric at $T_{\phi(w)}\Phi$ is $g_{ij,\phi(w)} = \mathbb{E}[\partial_{w^i} \log p(x,w) \cdot \partial_{w^j} \log p(x,w)]$, so the Fisher-Rao metric on $T_p\mathcal{P}$ is

$$g_p(h_1, h_2) = \mathbb{E}[h_1(\log p)h_2(\log p)] = \int_M \frac{h_1 h_2}{p^2} d\mu,$$

since $h(\log p) = (d/dt)|_{t=0} \log(p + th) = h/p$.

For the MSE loss on $\mathcal{P}$, $\ell_E(p) = \frac{1}{2} \sum_{i=1}^k |p(x_i) - y_i|^2$, as in the $L^2$ case we formally get

$$g_p(\nabla \ell_{E,p}, h) = d\ell_{E,P}(h) = \sum_{i=1}^k (p(x_i) - y_i)h(x_i)$$

$$= \int_M \sum_{i=1}^k (p(x_i) - y_i)\delta_i(h)$$

$$= g_p(p^2 \sum_{i=1}^k (p(x_i) - y_i)\delta_i, h).$$

Under the usual assumption that $\int_M \delta_i d\mu = 1$, the formal gradient is

$$\nabla \ell_{E,p} = p^2 \sum_{i=1}^{k} (p(x_i) - y_i)\delta_i - \sum_{i=1}^{k} p^2(x_i)(p(x_i) - y_i),$$

where the constant second term ensures $\int_M \nabla \ell_{E,p} = 0$ formally. Again, $\nabla \ell_{E,p}$ does not exist in $T_p\mathcal{P}$, since delta functions are not elements of $T_p\mathcal{P}$. Similarly, if we use a more general loss function $L(z, w)\colon \mathbb{R} \times \mathbb{R} \to \mathbb{R}$ and set $\ell\colon \mathcal{P} \to \mathbb{R}$ by $\ell(p) = \sum_i L(p(x_i), y_i)$, then the formal gradient is $p^2 \sum_i (\partial L/\partial x)|_{(x_i,y_i)}\delta_i$, which again does not exist in $T_p\mathcal{P}$.

## 4 Gradients in RKHS and Generalized Tangent Kernel

The non-existence of the gradient of empirical loss discussed in §3 motivates our study of natural gradient in an RKHS, which is detailed in §4.1. In particular, whenever we have a pointwise defined loss function on a Riemannian Hilbert space/manifold $\mathcal{M}$ of functions, the functional gradient exists on $\mathcal{M}$ iff each $T_f\mathcal{M}$, $f \in \mathcal{M}$, is an RKHS. For a finite dimensional parameterized function space $\Phi \subset \mathcal{M}$, a natural gradient on $\Phi$ (and thus on the parameter space) is then determined by a projected RKHS from $T_f\mathcal{M}$ onto $T_f\Phi$. Our study goes further along this line in §4.2, where we introduce the Generalized Tangent Kernel (GTK), the core concept of this work, and prove an appealing automatic orthonormality property of GTK (Theorem 4.2). The GTK framework provides a unified understanding of projected functional gradients, producing families of natural gradients that include both the standard gradient and the classical natural gradient. In fact, a finite dimensional submanifold $\Phi \subset \mathcal{M}$ has many choices of natural gradients, which are in one-to-one correspondence with choices of GTKs on each $T_f\Phi$. Results in this section have proofs in Appendix E and numerical examples in Appendix G.

### 4.1 Natural Gradient in RKHS

In this subsection, we study the gradient of a loss function in an RKHS, which by definition is a function space where the evaluation/delta functions $\delta_x$ are continuous.

Let $\mathcal{H} \subset \mathcal{M}(M, \mathbb{R}^m)$ be a vector-valued RKHS with kernel $K_\mathcal{H}$. The reproducing property of $\mathcal{H}$ (see Appendix B.3) gives

$$\langle K_\mathcal{H}(x, \cdot), h \rangle_\mathcal{H} = (\langle K_\mathcal{H}^1(x, \cdot), h^1 \rangle, \ldots, \langle K_\mathcal{H}^n(x, \cdot), h^n \rangle) = (h^1(x), \ldots, h^n(x)) = h(x), \ \forall h \in \mathcal{H}.$$

Then there is a "$\delta$-vector" in the dual space of $\mathcal{H}$ that evaluates each component of $h$ at point $x$, i.e.,

$$[\boldsymbol{\delta}_x(h)]^j = \delta_x^j(h^j) = \langle K_\mathcal{H}^j(x, \cdot), h^j \rangle_{\mathcal{H}_j} = h^j(x),$$

where $\mathcal{H}_j$ is the scalar-valued RKHS corresponding to the $j$-th component of $\mathcal{H}$. Thus $d\ell_{E,f}(h) = \langle \nabla \ell_{E,f}, h \rangle_\mathcal{H}$ defines $\nabla \ell_{E,f}$ as an element of $\mathcal{H}$. In contrast, $h \mapsto d\ell_{E,f}(h)$ is not a continuous linear functional on the function spaces discussed in §3, so the functional gradient does not exist.

The next Lemma puts (12) into the RKHS framework for more general loss functions.

**Lemma 4.1.** *For $L = L(z, w)\colon \mathbb{R}^m \times \mathbb{R}^m \to \mathbb{R}$ a differentiable function, the $\mathcal{H}$-gradient of the $L$-empirical loss function $\ell_L(f) = \sum_i L(f(x_i), y_i)$ on $\mathcal{H} \subset \mathcal{M}(M, \mathbb{R}^m)$ is given by*

$$\nabla \ell_{L,f} = \sum_i K_\mathcal{H}(x_i, \cdot)\nabla^E L_z(f(x_i), y_i), \tag{13}$$

*where $\nabla^E L_z$ is the Euclidean gradient of $L$ in the $z$ direction.*

If $\mathcal{H} = \mathcal{M}(M, \mathbb{R}^m)$ with a suitable inner product, we know we should project $\nabla \ell_{L,f}$ to the tangent space of $\Phi = \mathrm{Im}(\phi)$. Proposition 2.1 and Remark 2.1 provide two formulas, based on the pushforward basis $\{\frac{\partial \phi}{\partial w^i}\}$ and the $\mathcal{H}$-orthonormal basis $\{h_i\}$ of $T_f\Phi$, respectively. Applying them to (13) gives

$$P\nabla \ell_{L,f} = \widetilde{g}_\mathcal{H}^{k\ell} \sum_i \left( \nabla^E L_z(f(x_i), y_i) \cdot_E \left. \frac{\partial \phi}{\partial w^k} \right|_{x_i} \right) \frac{\partial \phi}{\partial w^\ell}, \tag{14}$$

where $\widetilde{g}_{\mathcal{H}}$ is the pullback to $\mathcal{W}$ of the inner product $\langle \cdot, \cdot \rangle_{\mathcal{H}}$ on $\mathcal{H}$, and

$$P\nabla \ell_{L,f} = \sum_i \left( \nabla^E L_z(f(x_i), y_i) \cdot_E h_\ell \big|_{x_i} \right) h_\ell. \tag{15}$$

For this RKHS $\mathcal{H}$, the "$\delta$-vector" exists on each tangent space $T_f\Phi$ and is given by the corresponding "projected" evaluation function,

$$\boldsymbol{\delta_x}(h) = \langle h, P_f K_{\mathcal{H}}(x, \cdot) \rangle_{T_f\Phi} := \langle h, K_f(x, \cdot) \rangle_{T_f\Phi}, \text{ for } h \in T_f\Phi,$$

where $K_f = P_f K_{\mathcal{H}}$ is the projected reproducing kernel. As above, $K_f$ can be written using either the pushforward basis $\{\frac{\partial \phi}{\partial w^i}\}$ or the orthonormal basis $\{h_i\}$ of $T_f\Phi$. Combining these results, we have the following kernel form of the projected gradient, as an equivalent alternative to (14) and (15):

**Proposition 4.1.** *For $\mathcal{H} = \mathcal{M}(M, \mathbb{R}^m)$ an RKHS with kernel $K_{\mathcal{H}}$ and $L = L(z, w) \colon \mathbb{R}^m \times \mathbb{R}^m \to \mathbb{R}$ a differentiable function, the projection of the $\mathcal{H}$-gradient of the $L$-empirical loss function $\ell_L(f) = \sum_i L(f(x_i), y_i)$ to $\Phi$ equals*

$$\nabla^\Phi \ell_{L,f} := P\nabla \ell_{L,f} = \sum_i K_f(x_i, \cdot)\nabla^E L_z(f(x_i), y_i), \tag{16}$$

*where $K_f$ is the projection kernel of $K_{\mathcal{H}}$ onto the RKHS $T_f\Phi$. $K_f$ can be represented by the pushforward basis $\{\frac{\partial \phi}{\partial w^i}\}$ of $T_f\Phi$,*

$$K_f(x, x') = \tilde{g}_{\mathcal{H}}^{ij} \frac{\partial \phi}{\partial w^i}(x) \otimes \frac{\partial \phi}{\partial w^j}(x'), \tag{17}$$

*or by an orthonormal basis $\{h_i\}$ of $T_f\Phi$,*

$$K_f(x, x') = \sum_i h_i(x) \otimes h_i(x'). \tag{18}$$

(Note that $T_f\Phi$ is an RKHS, as it is finite dimensional subspace of the RKHS $T_f\mathcal{H} \simeq \mathcal{H}$.) In other words, in RKHS theory the projection view of Figure 1 and §2.1 produces a *family* of projected RKHSs parametrized by $w \in \mathcal{W}$, with the corresponding projected reproducing kernel $K_f$, for $f = \phi(w)$.

From a practical perspective, although different choices of $\mathcal{H}$ produce different natural gradients, in general, it is expensive to compute either $\widetilde{g}_{\mathcal{H}}^{ij}$ in (17) or $\{h_i\}$ in (18). For $\mathcal{H} = H_s(\mathbb{R}^n, \mathbb{R}^m)$ a Sobolev space (detailed in Appendix B.2), an explicit computation of $K_{\mathcal{H}}(x, \cdot)$ and hence of $\boldsymbol{\delta_x}$ is in Rosenberg (2023). Because of this, we introduce a new variant of natural gradient, called the Sobolev natural gradient in Appendix G, where we discuss an RKHS-based approximation of $\widetilde{g}_{\mathcal{H}}^{ij}$ and provide a practical computational method of $\widetilde{g}_{\mathcal{H}}^{ij}$ based on the Kronecker-factored approximation (Martens & Grosse, 2015; Grosse & Martens, 2016). We also report preliminary experimental results on image classification benchmarks.

Note that the RKHS form of projected gradient (16), in particular the projected kernel forms (17) and (18), is consistent with our observations from Proposition 2.1 and Remark 2.1, namely that the inverse metric matrix of natural gradient appears as a consequence of orthogonal projection from $\mathcal{M}$ to $\Phi$. This matrix becomes an identity matrix iff an orthonormal basis on each tangent space of $\Phi$ is used.

## 4.2 A Unified Perspective with Generalized Tangent Kernel

In this section, we introduce a general form of the kernel (18), which provides a unified characterization as well as a new understanding of projected functional gradients, including both natural gradient and standard gradient. The favorable property of this kernel also enables us to address the non-immersion case of $\phi$ in §5.

From §2, the standard gradient formula (5) for the pushforward basis is similar to the natural gradient formula (4) for an orthonormal basis. This formal similarity extends to the corresponding RKHS expressions; as with (16) for the natural gradient, there is a kernel form for the standard gradient (5):

$$\nabla^\Phi \ell_{L,f} := S\nabla \ell_{L,f} := \sum_i \Theta(x_i, \cdot)\nabla^E L_z(f(x_i), y_i), \tag{19}$$

where $\Theta(x, x') = \sum_i \frac{\partial \phi}{\partial w^i}(x) \otimes \frac{\partial \phi}{\partial w^i}(x')$ happens to be the Neural Tangent Kernel (NTK) (Jacot et al., 2019). This observation motivates the following generalized form of both $K_f$ and $\Theta$.

**Definition 4.1** (Generalized Tangent Kernel). *For $\phi \colon \mathcal{W} \to \mathcal{M}$ an immersion, where $\mathcal{W}$ is a parameter space and $\mathcal{M}$ is a function space, let $\{b_i\}$ be a basis of the tangent space $T_f \Phi$ at $f = \phi(w)$ for some $w \in \mathcal{W}$. The Generalized Tangent Kernel of $T_f \Phi$ is defined to be*

$$K_{GTK}(x, x') = \sum_i b_i(x) \otimes b_i(x'). \tag{20}$$

The Generalized Tangent Kernel (GTK) thus provides a unified characterization of gradients on $T_f \Phi$, with natural gradient and standard gradient as special cases. The natural gradient (16) is most closely related to the functional gradient in $\mathcal{M}$ (w.r.t. some functional metric), as it is obtained by an orthogonal projection and the corresponding GTK (18) is defined by an orthonormal basis of $T_f \Phi$. The standard gradient (19), in contrast, corresponds to NTK, a GTK defined by the pushforward basis from $\mathcal{W}$, which in general is not orthonormal for the functional metric. Thus NTK and any GTK besides (18) lose extra information about the functional gradient in $\mathcal{M}$.

So far, our presentation of the role of the orthogonal projection in natural gradient provides an intuitive justification for the long-believed statement that, regardless of computational cost, natural gradient has an advantage over standard gradient in capturing the geometric structure of the immersed function space. With the following favorable property of GTK, however, we see the surprising result that the standard gradient is, at least locally, as geometrically informative as the natural gradient.

**Theorem 4.2** (**Automatic Orthonormality of GTK**). *Given a finite dimensional function space $\mathcal{F} \subset \{f \colon X \to \mathbb{R}^n\}$, where $X$ is any set, and any basis $\{b_i\}$ of $\mathcal{F}$, $\{b_i\}$ is an orthonormal basis of the RKHS associated with the reproducing kernel given by the Generalized Tangent Kernel (20).*

*Proof.* Given a basis $\{b_i\}$, let $\mathcal{H}_K$ denote the RKHS associated with the reproducing kernel $K$ defined by the GTK (20). Then each $b_i$ is in the span of $\{K(x, \cdot), x \in X\}$ (Manton & Amblard, 2015, Lemma 2.1), and can be respresented as $b_i(\cdot) = \sum_{x \in X} K(x, \cdot) C_i(x)$, for some function $C_i \colon X \to \mathbb{R}^n$. By (20),

$$b_i(\cdot) = \sum_{x \in X} C_i(x)^T \sum_\ell b_\ell(x) \otimes b_\ell(\cdot) = \sum_\ell \left( \sum_{x \in X} C_i(x)^T b_\ell(x) \right) b_\ell(\cdot) = \sum_\ell \alpha_i^\ell b_\ell(\cdot), \tag{21}$$

where $\alpha_i^\ell = \sum_{x \in X} C_i(x)^T b_\ell(x)$. Since $\{b_i\}$ is a basis, the representation (21) must be unique, so $\alpha_i^\ell = \delta_i^\ell$. Thus by (36),

$$\begin{aligned}
\langle b_i, b_j \rangle_{\mathcal{H}_K} &= \left\langle \sum_{x \in X} K(x, \cdot) C_i(x), \sum_{x' \in X} K(x', \cdot) C_j(x') \right\rangle_{\mathcal{H}_K} = \sum_{x \in X} \sum_{x' \in X} C_i(x)^T K(x, x') C_j(x') \\
&= \sum_{x \in X} \sum_{x' \in X} C_i(x)^T \sum_\ell b_\ell(x) \otimes b_\ell(x') C_j(x') = \sum_\ell \left( \sum_{x \in X} C_i(x)^T b_\ell(x) \right) \left( \sum_{x' \in X} C_j(x') b_\ell(x') \right) \\
&= \sum_\ell \alpha_i^\ell \alpha_j^\ell = \delta_j^i,
\end{aligned}$$

where $\delta_j^i$ equals 1 when $i = j$ and 0 otherwise. $\square$

Theorem 4.2 is the key result in this work, as its direct Corollary expands the scope of natural gradient in an unexpectedly flexible way.

**Corollary 4.3** (**Natural Gradient for Arbitrary Basis**). *For $f = \phi(w)$, any choice of the basis $\{b_i\}$ of $T_f \Phi$ produces an orthogonal projection onto $T_f \Phi$ and thus a well-defined natural gradient at $w$. The natural gradient has the following kernel form,*

$$\nabla^\Phi \ell_{L,f} := P \nabla \ell_{L,f} = \sum_i K_{GTK}(x_i, \cdot) \nabla^E L_z(f(x_i), y_i), \tag{22}$$

*where $K_{GTK}$ is defined by (20).*

As a special case of of Corollary 4.3, for $\{b_i\} = \{\frac{\partial \phi}{\partial w^i}\}$, GTK recovers NTK, and (22) recovers the standard gradient formula (19), which is equivalent to the conventional standard gradient on the parameter space $\mathcal{W}$:

$$\nabla_w \ell_{L,\phi(w)} := \sum_i \nabla^E L_z \left( \phi(w)(x_i), y_i \right) \nabla_w \phi(w)(x_i). \tag{23}$$

If $\phi$ is a neural network with parameters $w$, then $\nabla^E L_z$ is the gradient of the loss function with respect to the network output and $\nabla_w \phi$ is the gradient of the the network output with respect to the network parameters. (23) can be efficiently computed by backpropagation. Corollary 4.3 guarantees that the standard gradient (23) used everywhere in machine learning is, in fact, a well-defined natural gradient, in the sense that it is the projection (with respect to some specific Riemannian metric) of the functional gradient in the ambient function space.

To see the flexibility of Corollary 4.3 in defining valid natural gradients, we now give an example of changing a GTK by a change of coordinates on $\mathcal{W}$ (which corresponds to a change of Riemannian metric on $\Phi$). Let $(\xi^1, \ldots, \xi^n)$ be a new set of coordinates on the parameter space $\mathcal{W}$, given by a full-rank transformation matrix $A = (a_i^j)$, i.e., $\xi^j = a_i^j w^i$. Then $f = \phi(w) = \phi(A^{-1}\xi) = \tilde{\phi}(\xi)$ and $\frac{\partial \tilde{\phi}}{\partial \xi^i} = \frac{\partial \phi}{\partial w^j} \tilde{a}_i^j$, where $(\tilde{a}_i^j) = A^{-1}$. Instead of choosing the pushforward of $\{\frac{\partial}{\partial w^i}\}$ as basis, i.e., $\{b_i\} = \{\frac{\partial \phi}{\partial w^i}\}$, and getting the standard gradient (23), we choose the pushforward of $\{\frac{\partial}{\partial \xi^i}\}$, i.e., $\{b_i\} = \{\frac{\partial \tilde{\phi}}{\partial \xi^i}\}$, and apply Corollary 4.3 again. A change of coordinate computation gives the following altered gradient update rule on $\mathcal{W}$:

$$\begin{aligned}
\nabla_\xi \ell_{L,\tilde{\phi}(\xi)} &:= \sum_i \left( \nabla^E L_z \left( \tilde{\phi}(\xi)(x_i), y_i \right) \cdot_E \frac{\partial \tilde{\phi}}{\partial \xi^j}(x_i) \right) \frac{\partial \tilde{\phi}}{\partial \xi^j} \\
&= \sum_i \left( \nabla^E L_z \left( \tilde{\phi}(\xi)(x_i), y_i \right) \cdot_E \frac{\partial \phi}{\partial w^k}(x_i) \tilde{a}_j^k \right) \frac{\partial \phi}{\partial w^\ell} \tilde{a}_j^\ell \\
&= \sum_i \left( \nabla^E L_z \left( \phi(w)(x_i), y_i \right) \cdot_E \frac{\partial \phi}{\partial w^k}(x_i) \right) \tilde{a}_j^k \tilde{a}_j^\ell \frac{\partial \phi}{\partial w^\ell} \\
&= \sum_i \left( \nabla^E L_z \left( \phi(w)(x_i), y_i \right) \nabla_w \phi(w)(x_i) \right) A^{-1} \left( A^{-1} \right)^T \\
&= \underbrace{\left( \sum_i \nabla^E L_z \left( \phi(w)(x_i), y_i \right) \nabla_w \phi(w)(x_i) \right)}_{(23)} (A^T A)^{-1}.
\end{aligned} \tag{24}$$

We omit $\frac{\partial \phi}{\partial w^\ell}$ in the last two lines as in (23). (24) can also be efficiently computed by standard backpropagation, since the only extra term compared to (23) is the fixed $(A^T A)^{-1}$. In the next section, we further leverage the flexibility of Corollary 4.3 to propose new approaches to address the ill-defined natural gradients for non-immersion function approximations.

To the best of our knowledge, this is a new understanding of standard gradient, natural gradient, as well as NTK. It is also worth noting that while GTK is a generalized form of NTK, the theory developed in this section does not correspond to known results about NTK.

## 5  Natural Gradient for Non-Immersion Function Approximation

If $\phi$ is not an immersion, the pullback metric matrix $(\tilde{g}_{ij}) = \phi^* \bar{g}$ is singular. Therefore, the inverse matrix $(\tilde{g}^{ij})$ does not exist, and the formula (3) for natural gradient makes no sense. Neural networks with (piecewise) linear activations (Dinh et al., 2017) are such examples. Specifically, take a two layer neural network $\phi_\theta \colon \mathbb{R}^n \to \mathbb{R}^m$ with activation function $\psi$, so $\phi(\boldsymbol{\theta})(\boldsymbol{X}) = A_{2,\theta_1}(\psi(A_{1,\theta_2}\boldsymbol{X} + \boldsymbol{b}_{1,\theta_3}) + \boldsymbol{b}_{2,\theta_4}$, where $\boldsymbol{\theta} = (\theta_1, \theta_2, \theta_3, \theta_4)$ and $A_{2,\theta_1}$ has entries $\theta_1$, etc. Then $\phi(\theta_1, \theta_2, \theta_3, \theta_4) = \phi(\lambda^{-1}\theta_1, \lambda\theta_2, \lambda\theta_3, \theta_4)$ for any $\lambda \neq 0$.

Most existing solutions (Ollivier, 2015; Bernacchia et al., 2018) use the Moore-Penrose generalized inverse to define a natural gradient formula. However, the Moore-Penrose inverse in general loses more information

about the function space geometry and gradient than an orthogonal projection from $\mathcal{M}$ to $T_f\Phi$. Using Corollary 4.3, we propose an alternative approach in certain explicit cases, consisting of three steps:

1. For the given $\phi$, find a slice $\mathcal{S}$ (defined in §5.1) of the parameter space $\mathcal{W}$ with $\phi|_\mathcal{S}$ an immersion.

2. Compute a basis $\{c_i\}$ of each tangent space of $\mathcal{S}$.

3. Choose the basis $\{b_i\}$ on $T_f\Phi$ from the pushforward of $\{c_i\}$ and construct the corresponding GTK from $\{b_i\}$. By Corollary 4.3, (22) then gives a well-defined natural gradient.

This approach applies to any non-immersion function approximation. While finding the slice $\mathcal{S}$ may be difficult depending on the concrete function approximation, once $\mathcal{S}$ and its basis $\{c_i\}$ are computed, they can be used throughout the training.

This approach is also computationally efficient. Due to the flexibility of Corollary 4.3, we are allowed to choose the computationally simplest basis on $T_f\Phi$ to construct a GTK and (22) always gives a valid natural gradient. In the following section, we give two such examples for the ReLU MLP. Our experiments in Section 5.2 show that the extra computational cost compared to the standard gradient baseline[5] is negligible.

Note that a naive slice-based approach without leveraging Corollary 4.3 would make little sense in practice. In particular, the natural gradient (3) for the immersion case is known to be computationally expensive, and people rely on a series of approximations (Martens & Grosse, 2015; Grosse & Martens, 2016) to make it acceptable in practice. Restricting the natural gradient to a slice $\mathcal{S}$ is yet more complicated; in fact, it is unclear if a valid approximation still exists.

## 5.1 MLP with ReLU Activation

Following Dinh et al. (2017), we use ReLU MLP without bias terms as a concrete example to derive two variants of well-defined natural gradients. Proofs are in Appendix F, and results and proofs for a general ReLU MLP with bias terms are in Appendix I.

Our setup is a $\ell$-layer MLP with ReLU activations, with parameters $w_r^k$ in the $k$-th layer, for $r = 1, \ldots, n_k$, where $n_k$ is the number of matrix entries in the $k$-th layer. The space of parameter vectors is $\mathcal{W} = \{\boldsymbol{w} = (w_1^1, \ldots, w_{n_\ell}^\ell)\}$ and $\dim(\mathcal{W}) = \sum_\ell n_\ell$. The map $\phi\colon \mathcal{W} \to \mathrm{Maps}(\mathbb{R}^n, \mathbb{R}^m)$ takes a parameter vector to the associated MLP. We abbreviate the subvector for the $k$-th layer by $w^k = (w_1^k, \ldots, w_{n_k}^k)$, and $\boldsymbol{w} = (w^1, \ldots, w^\ell)$. We can assume that $w^k \neq \boldsymbol{0}$ for all $k$, since a MLP with some $w^k = \boldsymbol{0}$ is trivial.

Define a multiplicative group $G = \{\boldsymbol{\alpha} = (\alpha_1, \ldots, \alpha_\ell) : \alpha_i > 0, \prod_{i=1}^\ell \alpha_i = 1\}$. The map $G \to \mathbb{R}^\ell, (\alpha_1, \ldots, \alpha_\ell) \mapsto \sum_i \log \alpha_i$ is a bijection to the plane $\sum_i x^i = 0$, so $G$ is a manifold of dimension $\ell - 1$. $G$ acts on $\mathcal{W}$ by

$$(\alpha_1, \ldots, \alpha_\ell) \cdot (w^1, \ldots, w^\ell) = (\alpha_1 w^1, \ldots, \alpha_\ell w^\ell).$$

It is straightforward to check that

$$\phi(w^1, \ldots, w^\ell) = \phi(\boldsymbol{\alpha} \cdot (w^1, \ldots, w^\ell)), \tag{25}$$

for all $\boldsymbol{\alpha} \in G$. Thus $\phi$ is far from injective. As shown below, $\phi$ is not an immersion.

**Proposition 5.1.** *The dimension of $\ker(d\phi)_{\boldsymbol{w}}$ at each $\boldsymbol{w} \in \mathcal{W}$ is at least $\ell - 1$. Thus $\phi$ is not an immersion. In fact, the span of*

$$\left\{ Z^j|_{\boldsymbol{w}} = \sum_{r=1}^{n_1} w_r^1 \frac{\partial}{\partial w_r^1}\bigg|_{\boldsymbol{w}} - \sum_{s=1}^{n_j} w_s^j \frac{\partial}{\partial w_s^j}\bigg|_{\boldsymbol{w}} = \frac{\partial}{\partial r^1}\bigg|_{w^1} - \frac{\partial}{\partial r^j}\bigg|_{w^j} : j = 2, \ldots, \ell \right\} \tag{26}$$

*is in $\ker(d\phi)_{\boldsymbol{w}}$. Here $r^i$ is the radial vector in the parameter space of the $i^{\text{th}}$ layer.*

---

[5]Here we treat the standard gradient as if $\phi$ were an immersion, so strictly speaking Corollary 4.3 does not apply and this gradient is not a valid natural gradient.

**Remark 5.1.** *This issue is fundamentally caused by the homogeneity of a neural network activation function, as any (piecewise) linear activation function (ReLU, Leaky ReLU, Shifted ReLU, Parameterized ReLU, etc.) produces a non-immersive $\phi$.*

We now follow the three steps proposed in the previous section.

**Step 1.** Find a slice $\mathcal{S}$ of $\mathcal{W}$

Note that $\phi$ is constant along the orbits $\mathcal{O}_{\boldsymbol{w}} = \{g \cdot \boldsymbol{w} : g \in G\}$. Thus all the information in $\phi$ is contained in a slice for $G$, *i.e.,* a submanifold $\mathcal{S}$ of $\mathcal{W}$ that intersects each orbit once. The following theorem gives an example of $\mathcal{S}$.

**Theorem 5.2.** *Write $\mathcal{W} = \mathbb{R}^{n_1} \times \ldots \times \mathbb{R}^{n_\ell}$. A slice $\mathcal{S}$ is given by $\mathcal{S} = \{\lambda(w^1, \ldots, w^\ell) : w_i \in S^{n_i-1}, \lambda > 0\}$, where $S^{k-1}$ is the unit sphere in $\mathbb{R}^k$.*

For $\boldsymbol{w}^0 \in \mathcal{S}$, it is unlikely that any nonzero $v$ in $T_{\boldsymbol{w}^0}\mathcal{S}$ lies in $\ker(d\phi)_{\boldsymbol{w}^0}$. By a dimension count, $\dim(T_{\boldsymbol{w}^0}\mathcal{S}) + \dim(\mathrm{Span}\{Z^j\}) = \dim(\mathcal{W})$. Therefore, we expect $\ker(d\phi)_{\boldsymbol{w}^0} = \mathrm{Span}\{Z^j\}$ in Prop. 5.1, so that $\phi|_{\mathcal{S}}$ is immersion.

**Step 2.** Find a basis $\{c_i\}$ of $\mathcal{S}$.

We take a basis $\{\boldsymbol{r} := c_0^0, c_k^{k_i} : k = 1, \ldots, \ell; k_i = 1, \ldots, n_k - 1\}$ of the tangent space $T_{\boldsymbol{w}}\mathcal{S}$ for the slice $\mathcal{S}$. In polar coordinates $(r_k, \psi_k^{k_i}), k_i = 1, \ldots, n_k - 1$, on the $k^{\mathrm{th}}$ layer, $\boldsymbol{r} := \sum_k \partial/\partial r_k, c_k^{k_i} = \partial/\partial \psi_k^{k_i}$, at $\boldsymbol{x} \in \mathbb{R}^{n_k}$. The Euclidean metric restricts to a metric on $\mathcal{S}$ with metric tensor $h = h_{(k,k_i),(s,s_j)} = c_k^{k_i} \cdot c_s^{s_j}$. Details of $\{c_i\}$ and $h$ are given in Appendix F.

**Step 3.** Define a GTK.

Given the flexibility of Corollary 4.3, we suggest two approaches for constructing the GTK from pushforward of $\{c_i\}$.

*Approach I: NTK.* We push forward the metric $h$, the Euclidean inner product restricted to $\mathcal{S}$, to $T_f\Phi$. Thus we replace $\Theta$ in (19) with $\sum_{k,k_i} h^{(k,k_i),(r,r_j)} d\phi(c_k^{k_i}) \otimes d\phi(c_r^{r_i})$.

*Approach II: GTK.* We can directly use (20) to get $K_{GTK} = \sum_{k,k_i} d\phi(c_k^{k_i}) \otimes d\phi(c_k^{k_i})$.

For Approach I, the final expression for the natural gradient is given by the following proposition.

**Proposition 5.2.** *The gradient $\nabla^{\mathcal{S}}\widetilde{F}$, for $\widetilde{F} \colon \mathcal{W} \to \mathbb{R}$, is given by*

$$
\nabla^{\mathcal{S}}\widetilde{F}_{\boldsymbol{w}} = \sum_{k=1}^{\ell} \sum_{k_j=1}^{n_k} \frac{\partial \widetilde{F}}{\partial w_{k_j}^k} \frac{\partial}{\partial w_{k_j}^k} - \frac{1}{|w_1|^2} \sum_{k=1}^{\ell} \left( \sum_{k_i=1}^{n_k} w_{k_i}^k \frac{\partial \widetilde{F}}{\partial w_{k_i}^k} \right) \sum_{k_j=1}^{n_k} w_{k_j}^k \frac{\partial}{\partial w_{k_j}^k}
$$
$$
+ \frac{\ell^{-1}}{|w_1|^2} \left( \sum_{s=1}^{\ell} \sum_{s_i=1}^{n_s} w_{s_i}^s \frac{\partial \widetilde{F}}{\partial w_{s_i}^s} \right) \sum_{k=1}^{\ell} \sum_{k_j=1}^{n_k} w_{k_j}^k \frac{\partial}{\partial w_{k_j}^k}. \tag{27}
$$

Details of the notation and the proof are in Appendix F. Gradient formulas for Approach II are in Appendix I.2.

The procedure of training MLP with ReLU activations using Proposition 5.2 is summarized in Algorithm 1. (27) may seem complicated, but in fact, it can be easily implemented, and the extra computational cost versus standard gradient computation is negligible. Preliminary experimental results of both Approach I and II are provided in the next section and a PyTorch implementation of Approach I is included in Supplementary Materials.

## 5.2 Experiments

Following Algorithm 1, we implemented a stochastic version of the proposed natural gradient descent for a seven-layer MLP with ReLU activations. We test our implementation (slice SGD) versus standard SGD on the CIFAR-10 image classification benchmark. We use a seven-layer MLP of layer size $(2634, 2196, 1758, 1320, 882, 444, 10)$,

---

**Algorithm 1** Slice SGD for MLP with ReLU activations

---

**Require**: stepsize $\alpha$

**Init**: network weights $\boldsymbol{w}^0 = (w_{1,1}, \ldots, w_{\ell,n_\ell}) \in \mathcal{S}$, $t = 0$

   **while** stopping criterion not met **do**

      Compute Euclidean gradients $\left.\frac{\partial \widetilde{F}}{\partial w_{k,k_i}}\right|_{\boldsymbol{w}=\boldsymbol{w}^t}$   via standard backpropagation

      Compute $\nabla^{\mathcal{S}} \widetilde{F}_{\boldsymbol{w}^t}$ by (27)

      Update weights $\bar{\boldsymbol{w}}^{t+1} = \boldsymbol{w}^t + \alpha \cdot \nabla^{\mathcal{S}} \widetilde{F}_{\boldsymbol{w}^t}$

      Move $\bar{\boldsymbol{w}}^{t+1}$ along its G-orbit back into $\mathcal{S}$ by $\boldsymbol{w}^{t+1} = \left(\prod_{i=1}^{\ell} |\bar{w}_i^t|\right)^{1/\ell} \left(\frac{\bar{w}_1^t}{|\bar{w}_1^{t+1}|}, \ldots, \frac{\bar{w}_\ell^{t+1}}{|\bar{w}_\ell^{t+1}|}\right) \in \mathcal{S}$

      $t \leftarrow t + 1$

   **end while**

---

all layers except the last uses the ReLU activation function. This MLP is trained by standard SGD and our slice SGD methods using a batch size 128 for 200 epochs respectively. To simplify the comparison, we have not used any weight decay or momentum. The same fixed learning rate of 0.01 is used for all methods. Note that these basic settings are commonly used for the SGD baseline, which have been extensively optimized by the ML community. The learning rate is also tuned for the baseline. Our methods simply align everything with the baseline without specific tuning. All experiments are run on a desktop with an Intel i9-7960X 16-core CPU, 64GB memory, and a GeForce RTX 2080Ti GPU.

The testing performance vs wall-clock training time is shown in Figure 2(a). Though our proposed slice SGD formulas seem complicated, the implementation in PyTorch, as attached in Supplementary Materials, is straightforward. The extra computational cost compared to standard SGD is $O(n)$ scalar multiplications and additions, where $n$ is the number of network parameters. Note that these extra computations are between network weights and their gradients (already) obtained by standard backpropagation, which can be effectively leveraged by modern GPU's. This is fundamentally different from the sequential nature of backpropagation. As shown in the wall-clock axis of Figure 2(a), the difference in time cost versus standard SGD is negligible in practice. We have also tested Approach II described in §5.1, with the corresponding gradient formulas provided in Appendix I.2. As shown in Figure 2(b), it performs similarly to the Approach I.

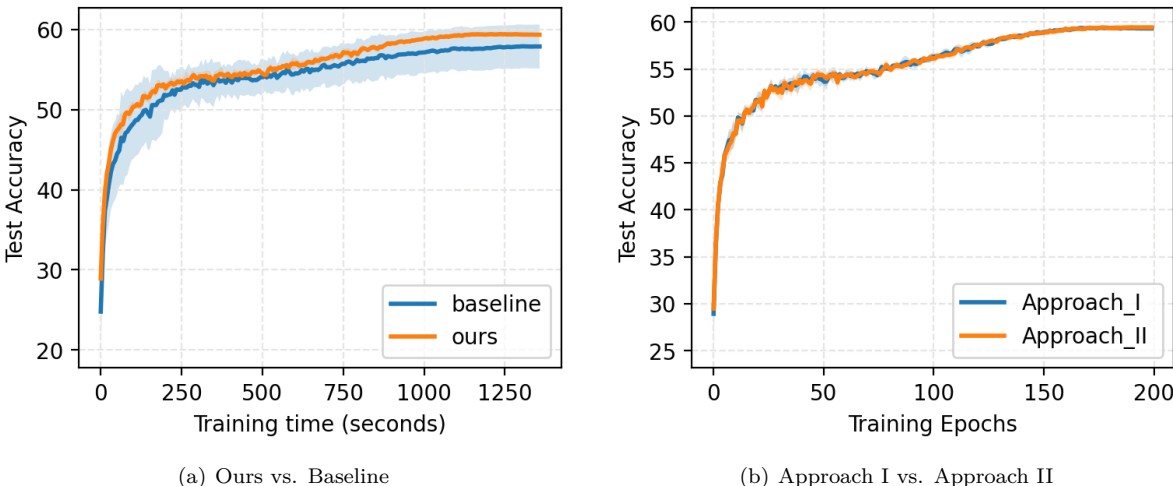

(a) Ours vs. Baseline            (b) Approach I vs. Approach II

Figure 2: (a) Testing accuracy vs. wall-clock of the slice SGD (ours) vs standard SGD (baseline) on CIFAR-10. (b) Testing accuracy vs. training epochs of Approach I and Approach II of our slice SGD on CIFAR-10. Results are averaged over five runs of different random seeds, with the shaded area corresponding to the standard deviation.

While our experimental results are still preliminary, we feel that they show the potential of our theory, while leaving a large room for practical optimization to future work.

## 6   Discussion

We believe our work is just the beginning of fruitful results in both theory and practice of a broader view of natural gradient. From a theoretical viewpoint, the Generalized Tangent Kernel, in particular its automatic orthonormality property, induces a family of Riemannian metrics on the entire function space $\mathcal{M}$, putting both the natural/pullback metric and the Euclidean/pushforward metric into a unified framework, as both metrics reflect (different) geometries on the function space. This motivates the question of which metrics in $\mathcal{M}$ lead to better convergence rate for natural gradient descent. Even on a finite dimensional manifold, the relation between a choice of metric and the convergence rate seems to be not well studied. As a first step towards comparing pushforward and pullback metrics, we compare notions of flatness in Appendix H.

There are many more topics in Riemannian geometry and machine learning to explore. For example, we have not discussed the relation between the choice of metric and the generalization ability. A related example is Kozachkov et al. (2023), which establishes an interesting connection between Riemannian contraction and an algorithmic stability generalization bound. In another direction, the very intriguing question of which flows on a manifold can be put into gradient flow form for some Riemannian metric (Shoji et al., 2024) is equivalent to determining which flows can be put into GTK form; this will be discussed in future work.

Transformers are known to possess the in-context learning capability. It has been shown (Ahn et al., 2023) that transformers actually learn to implement preconditioned gradient descent for the forward pass. Whether there is a GTK explanation for this preconditioned gradient descent in forward pass is another interesting question for future study.

From a practical viewpoint, our study and experiments of the new families of natural/standard gradient descent methods motivated by our theory are still preliminary. We also hope to study slices of parameter spaces for non-immersion function approximations beyond MLPs, such as convolutional neural networks, ResNets, and Transformers. It is therefore appealing from both theoretical and practical perspectives to see if a general approach to the slice SGD can be designed for different neural network architectures.

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

## A Key Notations

Table 1: Key Notations

| |
|---|
| $\mathcal{M} = \mathcal{M}(M, N)$: the space of differentiable functions from a manifold $M$ to a manifold $N$; usually $M = \mathbb{R}^n, N = \mathbb{R}^m$ |
| $\phi\colon \mathcal{W} \to \mathcal{M}$ : function approximation, i.e., the mapping from parameter space $\mathcal{W}$ to function space $\mathcal{M}$ |
| $\Phi = \phi(\mathcal{W}) \subset \mathcal{M}$ : the parameterized function space, i.e., the image of $\phi$ on $\mathcal{M}$ |
| $F\colon \mathcal{M} \to \mathbb{R}$ : the loss function on $\mathcal{M}$ |
| $\widetilde{F} = F \circ \phi\colon \mathcal{W} \to \mathbb{R}$ : the loss function on $\mathcal{W}$ from $F$ induced by $\phi$ |
| $\bar{g} = (\bar{g}_{ij})$ : the Riemannian metric on $\mathcal{M}$ |
| $\widetilde{g} = (\widetilde{g}_{ij}) = \phi^* \bar{g}$ : the pullback metric on $\mathcal{W}$ |
| $\ell_E$ : the empirical MSE loss on $\mathcal{M}$, given some finite training set |
| $\nabla^\Phi$: the orthogonal projection of the gradient on a function space or RKHS to $\Phi$ |
| $\mathcal{W} = \{\boldsymbol{w} = (w_1^1, \ldots, w_{n_\ell}^\ell)\}$: the parameter space for $\ell$-layer ReLU MLP, $n_k$ is the number of parameters of the $k$-th layer |
| $w^k = (w_1^k, \ldots, w_{n_k}^k)$ : the vector of parameters of the $k$-th layer; we assume that $w^k \neq \boldsymbol{0}$ |
| $G = \{\boldsymbol{\alpha} = (\alpha_1, \ldots, \alpha_\ell) : \alpha_i > 0, \prod_{i=1}^\ell \alpha_i = 1\}$: the multiplicative group acting on (MLP) $\mathcal{W}$ by $\boldsymbol{\alpha} \cdot \boldsymbol{w} = (\alpha_1 w^1, \ldots, \alpha_\ell w^\ell)$ |
| $\mathcal{O}_{\boldsymbol{w}} = \{g \cdot \boldsymbol{w} : g \in G\}$ : the orbit of the $\boldsymbol{w} \in \mathcal{W}$ under the group action; $\phi$ is constant on $\mathcal{O}_{\boldsymbol{w}}$ |
| $\mathcal{S} = \{\lambda(w^1, \ldots, w^\ell) : w^i \in S^{n_i-1}, \lambda > 0\}$: a slice in $\mathcal{W}$ for the group action, where $S^{k-1}$ is the unit sphere in $\mathbb{R}^k$ |
| $\mathcal{H} \subset \mathcal{M}(M, \mathbb{R}^m)$ : a vector-valued RKHS with kernel $K_{\mathcal{H}}$ |
| $\ell_L$ : the more general $L$-empirical loss on $\mathcal{M}$, where $L = L(z, w)\colon \mathbb{R}^m \times \mathbb{R}^m \to \mathbb{R}$ is differentiable |
| $\nabla^E L_z$ : the Euclidean gradient of $L(z, w)$ in the $z$ direction |
| $\widetilde{g}_{\mathcal{H}} = (\widetilde{g}_{\mathcal{H}, ij})$ : the pullback to $\mathcal{W}$ of the inner product $\langle \cdot, \cdot \rangle_{\mathcal{H}}$ on $\mathcal{H}$ |
| $K_f = P_f K_{\mathcal{H}}$ : the projected reproducing kernel of $K_{\mathcal{H}}$ on the tangent space $T_f \Phi$ |
| $\Theta(x, x') = \sum_i \frac{\partial \phi}{\partial w^i}(x) \otimes \frac{\partial \phi}{\partial w^i}(x')$ : the Neural Tangent Kernel (NTK) (Jacot et al., 2019) |
| $K_{GTK}(x, x') = \sum_i b_i(x) \otimes b_i(x')$ : the Generalized Tangent Kernel (GTK) |

# B    Background Material

Since our framework involves Riemannian manifolds, RKHS, and Sobolev spaces, we briefly review aspects of this material. General references are Lee (2013) for manifolds, Gilkey (1995) for Sobolev spaces on domains in $\mathbb{R}^n$ and on manifolds, and Berlinet & Thomas-Agnan (2004) or Manton & Amblard (2015) for RKHSs.

## B.1    Riemannian Manifolds

We assume the reader is familiar with the concepts of manifolds and local charts. To avoid technicalities, we assume all manifolds are smooth and are embedded in some $\mathbb{R}^N$. Let $M$ be an $m$-dimensional manifold, denoted $M^m$. At a point $p \in M^m$, the tangent space $T_pM \simeq \mathbb{R}^m$ is the collection of tangent vectors to smooth curves passing through $p$: $T_pM = \{(d/dt|_{t=0})\gamma(t)\}$, where $\gamma \colon (-\epsilon, \epsilon) \to M$ is a smooth map for some $\epsilon > 0$ with $\gamma(0) = p$.

Associated to a smooth map $f \colon M^m \to N^n$ between manifolds is its differential $df_p \colon T_pM \to T_{f(p)}N$, the best linear approximation to $f$ at $p$, defined by $df_p(X) = (d/dt|_{t=0})f(\gamma(t))$, where $\gamma(t)$ has $\gamma(0) = p, (d/dt|_{t=0})\gamma(t) = X$. (In many texts, $df_p$ is denoted by $f_{*,p}$.) We suggest the reader draw a picture for this definition. The differential is a linear transformation between tangent spaces; for $M = \mathbb{R}^m, N = \mathbb{R}^n$, the differential is the usual Jacobian matrix $df_p = (\partial f^i/\partial x^j|_p)_{n \times m}$. The equality $d(f \circ g) = df \circ dg$ for $f \colon M \to N$, $g \colon W \to M$ has a picture proof, and for $M, N, W$ Euclidean spaces is exactly the chain rule. As a special case, for $f \colon M \to \mathbb{R}$, $df_p \colon T_pM \to T_{f(p)}\mathbb{R} \simeq \mathbb{R}$ is an element of $T_p^*M$, the dual vector space of linear functionals on $T_pM$.

A manifold $M$ is a Riemannian manifold if it has a positive definite inner product $g_p$ on each $T_pM$, with the condition that $g_p$ depends smoothly on $p$ in a technical sense. Thus $g_p \colon T_pM \times T_pM \to \mathbb{R}$, with (i) $g(X, Y) = g(Y, X)$ and (ii) $g(X, X) \geq 0$ with $g(X, X) = 0$ iff $X = 0$. The dot product $X \cdot_m Y$ on $M = \mathbb{R}^m$ is the basic example of a Riemannian metric, and for $M \subset \mathbb{R}^N$, we can set $g_p(X, Y) = X \cdot_N Y$. However, $M$ has uncountably many Riemannian metrics; for example, we can take $h_p(X, Y) = A_p X \cdot_N Y$ for any positive definite symmetric matrix $A_p$ depending smoothly on $p$.

A finite dimensional vector space $V$ and its dual space $V^*$ of linear functionals $\lambda \colon V \to \mathbb{R}$ are isomorphic, just because they have the same dimension, but there is no canonical/natural isomorphism that doesn't involve choosing a basis of $V$. ("Canonical" means "without adding any more data.") In contrast, if $V$ has a positive definite inner product $\langle \cdot, \cdot \rangle$, then we have a canonical isomorphism $V \xrightarrow{\simeq} V^*$ given by $v \mapsto \lambda_v$, where $\lambda_v(w) = \langle v, w \rangle$ for $w \in V$. In particular, on a Riemannian manifold there is a canonical isomorphism between $T_pM$ and $T_p^*M$. Thus for a smooth function $f \colon M \to \mathbb{R}$, the differential $df_p \in T_p^*M$ corresponds to a unique tangent vector in $T_pM$, the gradient vector $\nabla f_p$. By the canonical isomorphism above, we have the fundamental formula

$$df_p(X) = g_p(\nabla f_p, X).$$

As expected, for $M = \mathbb{R}^m$, the gradient is the usual Euclidean gradient.

In Prop. 5.1, we will use the fact that for $f \colon M \to N$ and $n \in N$, $df|_{f^{-1}(n)} = 0$, i.e. the differential vanishes on the level set $f^{-1}(n)$: intuitively, $f$ does not change on the level set. (This assumes that the level set is a submanifold of $M$.) It follows that any tangent vector $v \in T_m f^{-1}(n)$ has $df(v) = 0$. Thus $T_m f^{-1}(n) = \mathrm{Ker}(df_m)$.

All this theory is useless unless we can do explicit calculations in local coordinates. If $\alpha_p \colon U_p \to \mathbb{R}^m$ is a local chart around $p \in M^m$, then $d\alpha_p \colon T_pU \to T_{\alpha(p)}\mathbb{R}^m \simeq \mathbb{R}^m$ is a vector space isomorphism. (Apply the chain rule to $d(\alpha^{-1} \circ \alpha) = d\mathrm{Id} = \mathrm{Id}$.) Thus the standard basis $\{e_1, \ldots, e_m\}$ of $\mathbb{R}^m$ corresponds to a basis of $T_pM$, confusingly denoted $\{\partial/\partial x^1|_p, \ldots, \partial/\partial x^m|_p\}$, or just $\{\partial_{x^1}|_p, \ldots, \partial_{x^m}|_p\}$. In this basis, the Riemannian metric has the local expression as a symmetric matrix:

$$g_p = (g_{ij})_p, \quad g_{ij} = g(\partial_{x^i}|_p, \partial_{x^j}|_p).$$

Dropping the $p$ index, if $X, Y \in T_pM$ are written as $X = X^i \partial_{x^i}$, $Y = Y^j \partial_{x^j}$ (using the Einstein summation convention that repeated indices are summed over), then $g(X, Y) = g_{ij} X^i Y^j$.

The corresponding dual basis of $T_p^*M$ is also confusingly denoted by $\{dx^1, \ldots, dx^m\}$, where $dx^i(\partial/\partial x^j) = \delta_j^i$, the Kronecker delta function. It follows that the local expression for $df_p$ is $df_p = (\partial f/\partial x^i)dx^i$. Thus the coordinate-independent differential $df_p$ keeps track of all first derivative information of $f$ in any local coordinate chart. The positive definite matrix $(g_{ij})$ is invertible, and its inverse is denoted by $(g^{ij})$. The canonical isomorphism $T_p^*M \to T_pM$ is given by $\beta = \beta_i dx^i \in T_p^*M \mapsto g^{ij}\beta_i\partial_{x^j} \in T_pM$. As a result, the local expression for the gradient of $f$ at $p$ is $\nabla f_p = g_p^{ij}(\partial f/\partial x^i)|_p\partial_{x^j}|_p$.

As a final technical point, a smooth function $\phi\colon M \to N$ pulls an inner product $g$ on $N$ back to an inner product denoted $\phi^*g$ on $M$ by setting

$$(\phi^*g)_p(X, Y) = g_{\phi(p)}(d\phi_p(X), d\phi_p(Y)). \tag{28}$$

However, a Riemannian metric may not pull back to a Riemmanian metric: if $X \neq 0$ but $d\phi(X) = 0$, then $(\phi^*g)(X, X) = 0$. Thus we must require that $\phi$ is an immersion, which by definition means that $d\phi_p$ is injective for all $p \in M$, in which case the *pullback* $\phi^*g$ of a Riemannian metric is a Riemannian metric.

## B.2 Sobolev Spaces

One motivation for Sobolev spaces comes from the fact that linear transformations on infinite dimensional normed vector spaces may not be continuous; this never happens in finite dimensional linear algebra. In fact, the simplest differential operator $d/dx$ on smooth functions on $[0, 1]$ does not extend to a continuous operator on the Hilbert space $L^2[0, 1]$: the functions $f_n(x) = n^{-1/2}\sin(2\pi nx)$ have $\lim_{n\to\infty}\|f_n\|_{L^2} = 0$, but $\lim_{n\to\infty}\|(d/dx)f_n\|_{L^2} = \infty$.

There are two approaches to handling differential operators as linear transformations on Banach/Hilbert spaces: (i) develop a theory of discontinuous/unbounded operators as in *e.g.*, Schmüdgen (2012); (ii) use Sobolev spaces to make the operators continuous. For (ii), if we define the first Sobolev space $H^1[0, 1]$ to be the Hilbert space completion of $C^\infty[0, 1]$ with respect to the norm $\|f\|_1 := \|f\|_{L^2} + \|(d/dx)f\|_{L^2}$, then it is immediate that $d/dx\colon H^1 \to L^2$ is continuous: $\lim_{n\to\infty}\|f_n\|_1 = 0$ implies $\lim_{n\to\infty}\|f_n\|_{L^2} = 0$.

In higher dimensions, let $\Omega$ be a bounded open set in $\mathbb{R}^n$. For $s$ a positive integer, we define the $s$-Sobolev space $H^s(\Omega)$ to be the Hilbert space completion of $C_c^\infty(\Omega, \mathbb{C})$ (smooth functions with compact support in $\Omega$) with respect to the norm

$$\|f\|_s^2 = \sum_{|\alpha|\leq s}\|\partial^\alpha f\|_{L^2}^2.$$

Here $\alpha = (\alpha^1, \ldots, \alpha^n)$ is a multi-index with $|\alpha| = \sum_i \alpha_i$, and $\partial^\alpha = \partial^{|\alpha|}/\partial x^{\alpha^1}\ldots\partial x^{\alpha^n}$. Thus the norm measures the $L^2$ norm of all partial derivatives of $f$ up to order $s$. By standard properties of the Fourier transform,

$$\|f\|_s^2 = \int_{\mathbb{R}^n}\sum_{|\alpha|\leq s}|\xi^\alpha|^2|\hat{f}(\xi)|^2 d\xi_1\ldots d\xi^n,$$

where $\xi = (\xi_1, \ldots, \xi_n)$ and $\xi^\alpha = xi_1^{\alpha_1}\cdot\ldots\cdot\xi_n^{\alpha_n}$. There are positive constants $C_1, C_2$ such that

$$C_1(1 + |\xi|^2)^s \leq \sum_{|\alpha|\leq s}|\xi^\alpha|^2 \leq C_2(1 + |\xi|^2)^s,$$

since these polynomials in the components of $\xi$ have the same degree, so the Sobolev $s$-norm is equivalent to the (renamed) norm

$$\|f\|_s^2 = \int_{\mathbb{R}^n}|\hat{f}(\xi)|^2(1 + |\xi|^2)^s d\xi_1...d\xi_n, \tag{29}$$

with the associated inner product $\langle f, g\rangle_s = \int_{\mathbb{R}^n}|\hat{f}(\xi)|\cdot|\overline{\hat{g}(\xi)}|(1 + |\xi|^2)^s d\xi_1...d\xi_n$. The advantage of (29) is that we can now define $H^s$ for any $s \in \mathbb{R}$, and we have the basic fact that there is a continuous nondegenerate pairing $H^s \otimes H^{-s} \to \mathbb{C}$, $f \otimes g \mapsto \int_\Omega f\cdot\overline{g}$. This implies that the dual space $(H^s)^*$ to $H^s$ is isomorphic to $H^{-s}$:

$$(H^{-s}) \xrightarrow{\simeq} (H^s)^* \text{ via } f \in H^{-s} \mapsto \left(g \in H^s \mapsto \int_\Omega f\cdot\overline{g}\right). \tag{30}$$

By the fundamental Sobolev Embedding Theorem, for $s > (n/2) + s'$, $H_s(\Omega)$ is continuously embedded in $C^{s'}(\Omega)$, the space of $s'$ times continuously differentiable functions on $\Omega$ with the sup norm. We always assume $s$ satisfies this lower bound for $s' = 0$. As a result, for any $x \in \Omega$, the delta function $\delta_x$ is in $H_s^*(\Omega)$: if $f_i \to f$ in $H^s$, then $f_i \to f$ in sup norm, so $\delta_x(f_i) = f_i(x) \to f(x) = \delta_x(f)$. Thus there exists $d_x \in H^s(M)$ such that

$$\delta_x(f) = \langle d_x, f \rangle_s, \forall f \in H^s. \tag{31}$$

All this extends to vector-valued functions. For $f = (f^1, ..., f^m) \in H^s(\Omega, \mathbb{R}^m)$, we can extend the inner product by

$$\langle f, g \rangle_{s^n} = \sum_j \langle f^j, g^j \rangle_s. \tag{32}$$

Then $\|f\|_s^2 = \sum_{j=1}^m \|f^j\|_s^2$, and we proceed as above to define $H_s(\Omega, \mathbb{R}^m)$. The delta function generalizes to $\boldsymbol{\delta}_x$, where

$$\boldsymbol{\delta}_x(f) = f(x), \ i.e., \ [\boldsymbol{\delta}_x(f)]^j = \langle d_x, f^j \rangle_s. \tag{33}$$

We also have Sobolev spaces $H_s(M)$ on manifolds $M$, which are defined first in local coordinates on each element in a coordinate cover of $M$, and then patched together using a partition of unity. Finally, we can combine these constructions to form $H_s(M, \mathbb{R}^m)$ for functions $f \colon M \to \mathbb{R}^m$.

## B.3  RKHS

The most common Hilbert space is an $L^2$ space, *e.g.,* $L^2[0,1]$, the space of real-valued functions defined on $[0, 1]$ which are $L^2$ with respect to *e.g.,* Lebesgue measure. One subtlety of $L^2$ spaces is that two functions $f_1, f_2$ define the same element in $L^2[0,1]$ if $f_1 = f_2$ except on a set $S$ of measure zero. As a result, the evaluation maps (often called delta functions) $\delta_x \colon L^2[0,1] \to \mathbb{R}$, $x \in [0,1]$, $\delta_x(f) = f(x)$, are not well-defined; if $x \in S$, then $\delta_x(f_1) \neq \delta_x(f_2)$. Even on well-defined functions, evaluation maps need not be continuous: for example, if $f_n(x) = x^n$, then $\lim_{n\to\infty} \|f_n\|_{L^2} = 0$, but $|\delta_1(f_n)| = 1 \not\to 0$. This is a major issue in §3.

The simplest definition of a Reproducing Kernel Hilbert Space (RKHS) is a Hilbert space $(\mathcal{H}, \langle \cdot, \cdot \rangle_{\mathcal{H}})$ of real-valued functions defined on a set $X$ such that the evaluation maps $\delta_x \colon \mathcal{H} \to \mathbb{R}$ are continuous for all $x \in X$. Our basic example of an RKHS is $H_s(\Omega)$, $s > n/2$, in the previous subsection.

As in (31), by the Riesz representation theorem, for each $x \in X$, there exists $d_x \in \mathcal{H}$ such that

$$f(x) = \delta_x(f) = \langle d_x, f \rangle_{\mathcal{H}}. \tag{34}$$

Note that this key equation fails if $\delta_x$ is not continuous. We usually write $d_x(y) = K_{\mathcal{H}}(x, y)$, so

$$f(x) = \langle K_{\mathcal{H}}(x, \cdot), f \rangle, \tag{35}$$

and call $K_{\mathcal{H}}$ the reproducing or Mercer kernel of the RKHS. It is elementary to show that $\mathcal{H}$ is an RKHS iff it has a *reproducing kernel*, a function $K_{\mathcal{H}} \colon X \times X \to \mathbb{R}$ with $K_{\mathcal{H}}(x, \cdot) \in \mathcal{H}$ for all $x \in X$ and with (35).

Note that

$$K_{\mathcal{H}}(x, y) = \langle K_{\mathcal{H}}(x, \cdot), K_{\mathcal{H}}(y, \cdot) \rangle_{\mathcal{H}}, \tag{36}$$

where we consider $K_{\mathcal{H}} \colon X \to \mathcal{H}$. More generally, consider a *feature map* $\phi \colon X \to \mathcal{H}'$ from $X$ to a Hilbert space $\mathcal{H}'$ and the associated *kernel function* $K(x, y) = \langle \phi(x), \phi(y) \rangle_{\mathcal{H}'}$. Then there is an RKHS $\mathcal{H}_K$ with reproducing kernel $K$. In fact, $\mathcal{H}_K$ is the Hilbert space completion of $\mathrm{Span}\{K(x, \cdot) : x \in X\}$. As a consistency check, if we start with an RKHS $\mathcal{H}$ with associated $K_{\mathcal{H}}$, and set $\phi(x) = K_{\mathcal{H}}(x, \cdot)$, we get $K = K_{\mathcal{H}}$ and $\mathcal{H}' = \mathcal{H}_K = \mathcal{H}$, so we recover the original RKHS.

As with Sobolev spaces, this theory extends to a Hilbert space $\mathcal{H}$ of vector-valued functions $f \colon X \to \mathbb{R}^m$. Let $\pi_i \colon \mathbb{R}^m \to \mathbb{R}$ be projection to the $i^{\text{th}}$ coordinate, and set $\mathcal{H}_i = \{\pi_i \circ f : f = (f^1, \ldots f^m) \in \mathcal{H}\}$. Assuming $\boldsymbol{\delta}_x$ in (33) is continuous, the Hilbert space $\mathcal{H}_i$ has a reproducing kernel $K_i$, and we obtain a reproducing kernel $K \colon X \times X \to \mathbb{R}^m$ by

$$\langle K(x, \cdot), f \rangle_{\mathcal{H}} = (\langle K_1(x, \cdot), f^1 \rangle_{\mathcal{H}_1}, \ldots, \langle K_n(x, \cdot), f^m \rangle_{\mathcal{H}_n}) = (f^1(x), \ldots, f^m(x)) = f(x). \tag{37}$$

**Notation:** If a kernel is written as as finite sum $K(x,y) = \sum_i a_i(x) \otimes b_i(y)$ for $a_i, b_i \in \mathcal{H}$, then $\langle K(x,\cdot), f(\cdot) \rangle_{\mathcal{H}}$ is by definition $\sum_i a_i(x) \langle b_i, f \rangle_{\mathcal{H}} \in \mathcal{H}$. If the sum is infinite, convergence issues must be treated.

## C  Proofs for §2

*Proof of Lemma 2.1.* (i) Take $X \in T_{\phi(w)}\Phi$ and a curve $\gamma(t) \subset \Phi$ with $\gamma(0) = \phi(w), \dot{\gamma}(0) = X$. Then

$$d(F|_\Phi)(X) = \left.\frac{d}{dt}\right|_{t=0} (F|_\Phi)(\gamma(t)) = \left.\frac{d}{dt}\right|_{t=0} (F)(\gamma(t)) = \langle \nabla F, X \rangle_{\phi(w)} = \langle P(\nabla F), X \rangle_{\phi(w)}.$$

Since $d(F|_\Phi)(X) = \langle \nabla(F|_\Phi), X \rangle$ determines $\nabla(F|_\Phi)$, we conclude that

$$P(\nabla F) = \nabla(F|_\Phi). \tag{38}$$

The result follows.

(ii) Take a gradient flow line $\gamma(t)$ for $\widetilde{F}$, so $\dot{\gamma}(t) = \nabla\widetilde{F}_{\gamma(t)}$. Then $\phi \circ \gamma$ is a gradient flow line of $F|_\Phi$ iff

$$d\phi(\dot{\gamma})(\gamma(t)) = (\phi \circ \gamma)^{\cdot}(t) = \nabla(F|_\Phi)_{(\phi \circ \gamma)(t)}.$$

Substituting in $\dot{\gamma}(t) = \nabla\widetilde{F}_{\gamma(t)}$, and using (38), we need to show that

$$d\phi(\nabla\widetilde{F}_w) = P(\nabla F_{\phi(w)}). \tag{39}$$

Take $\widetilde{X} \in T_w\mathcal{W}$. Because $\phi$ is an isometry from $(\mathcal{W}, \widetilde{g})$ to $(\Phi, \bar{g})$, we have for each $w$

$$\langle d\phi(\nabla\widetilde{F}), d\phi(\widetilde{X}) \rangle_{\bar{g}} = \langle \nabla\widetilde{F}, \widetilde{X} \rangle_{\widetilde{g}} = d\widetilde{F}(\widetilde{X}) = d(F \circ \phi)(\widetilde{X}) = dF \circ (d\phi(\widetilde{X})) = \langle \nabla F, d\phi(\widetilde{X}) \rangle_{\bar{g}}$$
$$= \langle P(\nabla F), d\phi(\widetilde{X}) \rangle_{\bar{g}}.$$

Since $d\phi(T_w\mathcal{W})$ spans $T_{\phi(w)}\Phi$, this proves (39) and $d\phi(\nabla\widetilde{F}) = \nabla F$. $\quad\square$

*Proof of Proposition 2.1.* In general, we can compute the projection $P$ in a vector space as the solution to the minimization problem: the projection $P\boldsymbol{v}$ of a vector $\boldsymbol{v}$ onto a subspace $B$ is

$$P\boldsymbol{v} = \mathrm{argmin}_{\boldsymbol{y} \in B} \|\boldsymbol{v} - \boldsymbol{y}\|^2.$$

If $\{b_j\}$ is a basis of $B$, we want $\mathrm{argmin}_{a^i \in \mathbb{R}} \|\boldsymbol{v} - \sum a^i b_i\|^2$. Taking first derivatives of $\|\boldsymbol{v} - \sum a^i b_i\|^2$ with respect to $a^i$, we get

$$0 = -2\langle v, b_i \rangle + 2 \sum_j a^j \langle b_i, b_j \rangle.$$

For $\widetilde{g} = (\widetilde{g}_{ij}) = (\langle b_i, b_j \rangle)$, we get $a^j = \widetilde{g}^{ij}\langle \boldsymbol{v}, b_i \rangle$ and thus $P\boldsymbol{v} = \widetilde{g}^{ij}\langle \boldsymbol{v}, b_i \rangle b_j$, where $(\widetilde{g}^{ij}) = (\widetilde{g}_{ij})^{-1}$. For $\boldsymbol{v} = \nabla F, b_i = \partial\phi/\partial w^i$, we recover (3). $\quad\square$

## D  Technicalities Ignored in §3

In the formal computation of §3, for the sake of clarity in presentation, we ignore several nontrivial technical questions.

Firstly, we ignore the nontrivial question of the norm topologies on $\mathcal{M}$ and $T_f\mathcal{M}$. Instead, we formally assume an easy topology induced by the $L^2$ inner product, as this is common in the machine learning literature. In this topology, the gradient for the MSE loss function does not exist. We use abstract RKHS theory as a setting where this gradient exists, with a Sobolev topology for large enough Sobolev parameter as a concrete example. This highlights the fundamental issue that norm topologies on infinite dimensional vector spaces can be inequivalent, unlike in finite dimensions.

Secondly, the differential-gradient formula (11),

$$d\ell_{E,f}(h) = \langle \nabla \ell_{E,f}, h \rangle_{L^2} = \int_M \sum_j (\nabla \ell_{E,f})^j \cdot h^j,$$

is again formal, since a linear functional $\ell: \mathcal{H} \to \mathbb{R}$ on a Hilbert space $\mathcal{H}$ satisfies $\ell(h) = \langle w, h \rangle$ for some $w \in \mathcal{H}$ iff $\ell$ is continuous. Since we haven't specified the topology on $\mathcal{M}$, we can't discuss the continuity of $d\ell_E$. As in Remark 3.1, for $\ell$ a discrete loss function like MSE, $\ell$ is not continuous on $L^2$ but is continuous on high Sobolev spaces.

## E   Proofs for §4

*Proof of Lemma 4.1.* In the notation in §B.2, we have

$$\langle \nabla \ell_{L,f}, h \rangle_{s^n} = d\ell_{L,f}(h) = \frac{d}{dt}\Big|_{t=0} L(f_t(x_i), y_i) = \sum_{i,j} \left( \frac{\partial L}{\partial z^j}(f(x_i), y_i) \right) \cdot h^j(x_i). \tag{40}$$

For $\nabla^E L_z$ the Euclidean gradient of $L$ with respect to $z$ only, we have

$$\sum_j \left( \frac{\partial L}{\partial z^j}(f(x_i), y_i) \right) \cdot h^j(x_i) = \sum_j (\nabla^E L_z(f(x), y_i))^j \cdot \langle K_{\mathcal{H}}^j(x_i, \cdot), h^j \rangle_s$$

$$= \langle K_{\mathcal{H}}(x_i, \cdot)(\nabla^E L_z(f(x), y_i)), h \rangle_{s^n}.$$

$\square$

*Proof of Proposition 4.1.* The projection of $K_{\mathcal{H}}$ is given by

$$K_f(x, x') = \langle K_{\mathcal{H}}(x, \cdot), h_i \rangle h_i(x') = \sum_i h_i(x) \otimes h_i(x'),$$

which proves (18).

For (17), write $b_i = a_i^j \partial/\partial w^j$, $b_k = a_k^\ell \partial/\partial w^\ell$. Using the trace map $a \otimes b \mapsto \langle a, b \rangle_{T_f \Phi}$, we get

$$\delta_{ik} = \langle b_i, b_k \rangle_{T_f \Phi} = a_i^j a_k^\ell \langle \partial/\partial w^j, \partial/\partial w^\ell \rangle_{T_f \Phi} = a_i^j a_i^\ell \widetilde{g}_{j\ell}.$$

Thus $a_i^j a_i^\ell = \widetilde{g}^{j\ell}$, and $K_f(x, x') = \sum_i b_i(x) \otimes b_i(x') = \sum_i a_i^j \partial/\partial w^j \otimes a_i^\ell \partial/\partial w^\ell = \widetilde{g}^{j\ell} \partial/\partial w^j \otimes \partial/\partial w^\ell.$ $\square$

*Proof of Corollary 4.3.*

To produce an orthogonal projection and thus a well-defined natural gradient from an arbitrary basis $\{b_i\}$ on $T_f \Phi$, there are two steps.

The first step is making $\{b_i\}$ an orthonormal basis of the RKHS $T_f \Phi$, i.e.,

$$\langle b_i, b_j \rangle_{K_{GTK}} = \delta_j^i, \tag{41}$$

which is guaranteed by Theorem 4.2.

The second step is to define a normal space $\mathcal{N}_f$ to $T_f \Phi$ inside $T_f \mathcal{M}$, so that any $v \in T_f \mathcal{M}$ can be decomposed into tangential and normal components, i.e.,

$$v = v^T + v^{\mathcal{N}}, \quad \text{where } v^T \in T_f \Phi, \ v^{\mathcal{N}} \in \mathcal{N}_f.$$

Since $T_f \mathcal{M}$ comes with a Riemannian metric $\bar{g}$ (*e.g.*, a Sobolev metric), the easiest choice for the normal space is $\mathcal{N}_f = \{v \in T_f \mathcal{M} : \langle v, v' \rangle_{\bar{g}} = 0, \ \forall v' \in T_f \Phi\}$. Now we have a *mixed* metric on a tubular neighborhood of $\Phi$ in $\mathcal{M}$, which is given by (41) in directions tangent to $\Phi$ and by $\bar{g}$ in $\bar{g}$-normal directions. We then extend this metric to all of $\mathcal{M}$ by a partition of unity. By Proposition 4.1, the natural/pullback gradient under this mixed metric is given by the following kernel form,

$$\nabla^\Phi \ell_{L,f} := P \nabla \ell_{L,f} = \sum_i K_{GTK}(x_i, \cdot) \nabla^E L_z(f(x_i), y_i).$$

$\square$

# F Proofs for §5

## F.1 Proof of Proposition 5.1

(25) is equivalent to

$$\phi(w^1, \ldots, \lambda w^i, \ldots, \lambda^{-1} w^j, \ldots, w^\ell) = \phi(w^1, \ldots, w^i, \ldots, w^j, \ldots, w^\ell). \tag{42}$$

Note that $\lambda\lambda^{-1} = 1$, so this is the two-layer analogue of $\prod \alpha_i = 1$.

Differentiate (42) at a fixed $\boldsymbol{w}^0$ for $i = 1$ by taking $(d/d\lambda)|_{\lambda=1}$ of both sides. This yields

$$\sum_{r=1}^{n_1} \left.\frac{\partial\phi}{\partial w_r^1}\right|_{\boldsymbol{w}^0} w_r^{0,1} - \sum_{s=1}^{n_j} \left.\frac{\partial\phi}{\partial w_s^j}\right|_{\boldsymbol{w}^0} w_s^{0,j} = 0 \in \mathbb{R}^m. \tag{43}$$

(43) is equivalent to

$$d\phi_{\boldsymbol{w}^0}\left(\sum_{r=1}^{n_1} w_r^{0,1}\frac{\partial}{\partial w_r^1}\right) - d\phi_{\boldsymbol{w}^0}\left(\sum_{s=1}^{n_j} w_s^{0,j}\frac{\partial}{\partial w_s^j}\right) = 0. \tag{44}$$

Thus the $\ell - 1$ vectors

$$\left\{ Z^j|_{\boldsymbol{w}^0} = \sum_{r=1}^{n_1} w_r^{0,1}\frac{\partial}{\partial w_r^1} - \sum_{s=1}^{n_j} w_s^{j,0}\frac{\partial}{\partial w_s^j} : j = 2, \ldots, \ell \right\} \tag{45}$$

are in $\ker(d\phi)_{\boldsymbol{w}^0}$. Since $w_j^0 \neq 0$ for all $j$, these vectors are nonzero. They are clearly linearly independent, since only the $k$-th vector involves the $k$-th layer. Thus the span of the $Z^j$ is contained in $\ker(d\phi)_{\boldsymbol{w}^0}$. $\qquad\square$

*Proof of Theorem 5.2.* We first show that every element $\boldsymbol{w} = (w^1, \ldots, w^\ell) \in \mathcal{W}$ lies on the orbit through some slice element.

Let $\boldsymbol{\alpha} = (\alpha_1, \ldots, \alpha_\ell)$, with

$$\alpha_j = \left(\prod_{i=1}^\ell |w^i|\right)^{1/\ell} |w^j|^{-1}, \quad j = 1, \ldots, \ell.$$

Then $\boldsymbol{\alpha} \in G$ and $\boldsymbol{\alpha} \cdot \boldsymbol{w} \in \mathcal{S}$.

We now show that each orbit of $G$ meets $\mathcal{S}$ at exactly one point. If $\boldsymbol{\alpha}^1 \cdot \boldsymbol{w}_1 = \boldsymbol{\alpha}^2 \cdot \boldsymbol{w}_2$, then in the obvious notation,

$$\alpha_k^1\lambda^1 = \alpha_k^1|w_1^k| = \alpha_k^2|w_2^k| = \alpha_k^2\lambda^2 \text{ for all } k,$$

which implies

$$(\lambda^1)^\ell = \prod_k \alpha_k^1\lambda^1 = \prod_k \alpha_k^2\lambda^2 = (\lambda^2)^\ell.$$

Thus $|w_1^k| = \lambda^1 = \lambda^2 = |w_2^k|$ for all $k$, so $\alpha_k^1 = \alpha_k^2$. Since $w_1^k, w_2^k$ are on the same ray, they must be equal. Therefore, $\boldsymbol{w}_1 = \boldsymbol{w}_2, \boldsymbol{\alpha}^1 = \boldsymbol{\alpha}^2$. $\qquad\square$

## F.2 Details of the basis and the restricted metric on slice $\mathcal{S}$

We take a basis $\{\boldsymbol{r} := c_0^0, c_k^{k_i} : k = 1, \ldots, \ell; k_i = 1, \ldots, n_k - 1\}$ of the tangent space $T_{\boldsymbol{w}}\mathcal{S}$ for the slice $\mathcal{S}$. In polar coordinates $(r_k, \psi_k^{k_i}), k_i = 1, \ldots, n_k - 1$, on the $k^{\text{th}}$ layer, $\boldsymbol{r} := \sum_k \partial/\partial r_k, c_k^{k_i} = \partial/\partial\psi_k^{k_i}$, at $\boldsymbol{x} \in \mathbb{R}^{n_k}$. The Euclidean metric restricts to a metric on $\mathcal{S}$ with metric tensor $h = h_{(k,k_i),(s,s_j)} = c_k^{k_i} \cdot c_s^{s_j}$.

On $\mathbb{R}^n = (w^1, \ldots, w^n)$, introduce spherical coordinates $(r, \psi^1, \ldots, \psi^{n-1})$ by

$$w^n = r\cos(\psi^{n-1}), \tag{46}$$

$$w^k = r\left(\prod_{s=k}^{n-1}\sin(\psi^s)\right)\cos(\psi^{k-1}), \ \ k = 2, \ldots, n-1,$$

$$w^1 = r\prod_{s=1}^{n-1}\sin(\psi^s).$$

Thus $\psi^{n-1} \in [0, \pi]$ is the pitch angle from $(w^1, \ldots, w^n)$ to the $w^n$-axis, $\psi^{n-2} \in [0, \pi]$ is the pitch angle from $(w^1, \ldots, w^{n-1}, 0)$ to the $w^{n-1}$ axis, etc., down to $\psi^2 \in [0, \pi]$. $\psi^1 \in [0, 2\pi]$ is the clockwise angle to the $w^2$-axis. (If the usual polar angle in the plane is $\theta$, then $\theta + \psi^2 = \pi/2$.) Using

$$\frac{\partial}{\partial\psi^\ell} = \frac{\partial w^k}{\partial\psi^\ell}\frac{\partial}{\partial w^k}, \frac{\partial}{\partial r} = \frac{\partial w^k}{\partial r}\frac{\partial}{\partial w^k},$$

we get

$$\partial_r = \frac{\partial}{\partial r} = \left(\prod_{s=1}^{n-1}\sin(\psi^s)\right)\partial_{w^1} + \sum_{\ell=2}^{n-1}\left(\prod_{s=\ell}^{n-1}\sin(\psi^s)^\prime\right)\cos(\psi^{\ell-1})\partial_{w^\ell} + \cos(\psi^{n-1})\partial_{w^n}$$

$$\partial_{\psi^1} = r\left(\prod_{s=2}^{n-1}\sin(\psi^s)\right)\cos(\psi^1)\partial_{w^1} - r\left(\prod_{s=1}^{n-1}\sin(\psi^s)\right)\partial_{w^2}$$

$$\partial_{\psi^\ell} = r\left(\prod_{s=\ell+1}^{n-1}\sin(\psi^s)\right)\cos(\psi^\ell)\left(\prod_{s=1}^{\ell-1}\sin(\psi^s)\right)\partial_{w^1}$$

$$+ r\sum_{k=2}^{\ell}\cos(\psi^\ell)\cos(\psi^{k-1})\left(\prod_{s=k}^{\ell-1}\sin(\psi^s)\right)\cdot\left(\prod_{s=\ell+1}^{n-1}\sin(\psi^s)\right)\partial_{w^k} \tag{47}$$

$$- r\left(\prod_{s=\ell}^{n-1}\sin(\psi^s)\right)\partial_{w^{\ell+1}}, \ \ \ell = 2, \ldots, n-2,$$

$$\partial_{\psi^{n-1}} = r\cos(\psi^{n-1})\left(\prod_{s=1}^{n-2}\sin(\psi^s)\right)\partial_{w^1} + r\sum_{k=2}^{n-2}\cos(\psi^{n-1})\cos(\psi^{k-1})\cdot\left(\prod_{s=k}^{n-2}\sin(\psi^s)\right)\partial_{w^k}$$

$$+ r\cos(\psi^{n-1})\cos(\psi^{n-2})\partial_{w^{n-1}} - r\sin(\psi^{n-1})\partial_{w^n}.$$

From (47), it is straightforward to check that

$$\langle\partial_r, \partial_{\psi^i}\rangle = \langle\partial_{\psi^i}, \partial_{\psi^k}\rangle = 0, \text{for } i \neq k, \langle\partial_r, \partial_r\rangle = 1,$$

$$\langle\partial_{\psi^i}, \partial_{\psi^i}\rangle = r^2\prod_{s=i+1}^{n-1}\sin^2(\psi^s). \tag{48}$$

Then the Euclidean metric/dot product restricts to a metric on $S$ with metric tensor

$$h = h_{(k,k_i),(s,s_j)} = c_k^{k_i} \cdot c_s^{s_j} = \begin{cases} 0, & k \neq s, \\ 0, & k = s \neq 0, k_i \neq s_j, \\ |w_k|^2\prod_{t=k_i+1}^{n_k-1}\sin^2(\psi_k^t), & k = s \neq 0, k_i = s_j, \\ \ell, & k = s = 0, i = j = 0. \end{cases} \tag{49}$$

Here the metric is computed at $\boldsymbol{w} = (w_1, \ldots, w_\ell) \in \mathcal{S}$ with $w_k = (r_k, \psi_k^i)$ the spherical coordinates of $\boldsymbol{w}$ in the $k^{\text{th}}$ layer. Note that $|w_k|^2 = r_k^2$ independent of $k$. $h$ is diagonal, so $h^{-1}$ is easy to compute.

For computational purposes, we need the right hand side of (47) in Euclidean coordinates. Set

$$a_\ell = \left(\sum_{i=1}^{\ell}(w^i)^2\right)^{1/2},$$

so in particular $a_n = r$. We have

$$\cos(\psi^{n-1}) = \frac{w^n}{a_n}, \quad \sin(\psi^{n-1}) = \frac{\left(a_n^2 - (w^n)^2\right)^{1/2}}{a_n}$$

$$\cos(\psi^\ell) = \frac{w^{\ell+1}}{a_n \prod_{s=\ell+1}^{n-1} \sin(\psi^s)}, \quad \ell = 1, \dots, n-2 \tag{50}$$

$$\sin(\psi^\ell) = (1 - \cos^2(\psi^\ell))^{1/2} = \frac{\left(a_n^2 \prod_{s=\ell+1}^{n-1} \sin^2(\psi^s) - \left(w^{\ell+1}\right)^2\right)^{1/2}}{a_n \prod_{s=\ell+1}^{n-1} \sin(\psi^s)}$$

Using (50), a downward induction from $\ell = n-1$ to 1 gives

$$\cos(\psi^\ell) = \frac{w^{\ell+1}}{a_{\ell+1}}, \quad \sin(\psi^\ell) = \frac{a_\ell}{a_{\ell+1}}, \quad \ell = 1, \dots, n-1. \tag{51}$$

Plugging (51 into (47) gives expressions for $\partial_r, \partial_{\psi^i}$ in Euclidean coordinates. In particular, in (49) we have the simplification

$$r^2 \sin^2(\psi_k^{n_k-1}) \cdot \dots \cdot \sin^2(\psi_k^{k_i+1}) = a_{k_i+1}^2. \tag{52}$$

We can also rewrite (47) as

$$\partial_r = \frac{1}{\sqrt{(w^1)^2 + \dots (w^n)^2}} \sum_{k=1}^n w^k \partial_{w^k},$$

$$\partial_{\psi^1} = w^2 \partial_{w^1} - w^1 \partial_{w^2}, \tag{53}$$

$$\partial_{\psi^\ell} = \sum_{k=1}^\ell \frac{w^k w^{\ell+1}}{\sqrt{(w^1)^2 + \dots + (w^\ell)^2}} \partial_{w^k} - \sqrt{(w^1)^2 + \dots + (w^\ell)^2} \partial_{w^{\ell+1}}, \quad \ell = 2, \dots, n-1.$$

### F.3 Proof of Proposition 5.2

For $\boldsymbol{w}$ in the slice $\mathcal{S}$, write $\boldsymbol{w} = (w^1, \dots, w^\ell)$ with $w^k = (w_1^k, \dots, w_{n_k}^k)$. Since the gradient is independent of coordinates, on the $k^{\text{th}}$ layer, $k = 1, \dots, \ell-1$, for any $\widetilde{F} \colon \mathcal{W} \to \mathbb{R}$ and any $\boldsymbol{w} = (w^1, \dots, w^k) \in \mathcal{W}$, we have

$$\nabla^{Euc,k} \widetilde{F}_{w^k} = \sum_{k_i=1}^{n_k} \frac{\partial \widetilde{F}}{\partial w_{k_i}^k} \frac{\partial}{\partial w_{k_i}^k}$$

$$= h_k^{0,0} \frac{\partial \widetilde{F}}{\partial r_k} \frac{\partial}{\partial r_k} + h^{0,k_i} \frac{\partial \widetilde{F}}{\partial r_k} \frac{\partial}{\partial \psi_k^{k_i}} + h^{k_i,0} \frac{\partial \widetilde{F}}{\partial \psi_k^{k_i}} \frac{\partial}{\partial r_k} + h^{k_i k_j} \frac{\partial \widetilde{F}}{\partial \psi_k^{k_i}} \frac{\partial}{\partial \psi_k^{k_j}}$$

$$= \frac{\partial \widetilde{F}}{\partial r_k} \frac{\partial}{\partial r_k} + \sum_{k_i=1}^{n_k-1} \left(\sum_{b=1}^{k_i+1} (w_b^k)^2\right)^{-1} \frac{\partial \widetilde{F}}{\partial \psi_k^{k_i}} \frac{\partial}{\partial \psi_k^{k_i}}.$$

The right hand side is evaluated at $w_k$. Here $h_k^{0,0} := \langle \partial_{r_k}, \partial_{r_k} \rangle^{-1} = 1$, we abbreviate $h^{(k,k_i),(k,k_j)}$ by $h^{k_i,k_j}$ when the context is clear, and we have used (52). For $\boldsymbol{w}$ in the slice $\mathcal{S}$, write $\boldsymbol{w} = (w^1, \dots, w^\ell)$ with

$w^k = (w_1^k, \ldots, w_{n_k}^k)$. Using a mixture of ordinary summation and summation convention, we have

$$
\begin{aligned}
\nabla^{\mathcal{S}} \widetilde{F}_{\boldsymbol{w}} &= h^{0,0} \frac{\partial \widetilde{F}}{\partial \boldsymbol{r}} \frac{\partial}{\partial \boldsymbol{r}} + \sum_{k=1}^{\ell} h^{k_i k_j} \frac{\partial \widetilde{F}}{\partial \psi_k^{k_i}} \frac{\partial}{\partial \psi_k^{k_j}} \\
&= \ell^{-1} \left( \sum_{k=1}^{\ell} \frac{\partial \widetilde{F}}{\partial r_k} \right) \left( \sum_{s=1}^{\ell} \frac{\partial}{\partial r_s} \right) + \sum_{k=1}^{\ell} \sum_{k_i=1}^{n_k-1} \left( \sum_{b=1}^{k_i+1} (w_b^k)^2 \right)^{-1} \frac{\partial \widetilde{F}}{\partial \psi_k^{k_i}} \frac{\partial}{\partial \psi_k^{k_i}} \\
&= \nabla^{Euc,\mathcal{W}} \widetilde{F} - \sum_{k=1}^{\ell} \frac{\partial \widetilde{F}}{\partial r_k} \frac{\partial}{\partial r_k} + \ell^{-1} \sum_{k=1}^{\ell} \sum_{s=1}^{\ell} \frac{\partial \widetilde{F}}{\partial r_k} \frac{\partial}{\partial r_s} \\
&= \sum_{k=1}^{\ell} \sum_{k_j=1}^{n_k} \frac{\partial \widetilde{F}}{\partial w_{k_j}^k} \frac{\partial}{\partial w_{k_j}^k} - \sum_{k=1}^{\ell} \sum_{k_i,k_j=1}^{n_k} \frac{w_{k_i}^k w_{k_j}^k}{|w^k|^2} \frac{\partial \widetilde{F}}{\partial w_{k_i}^k} \frac{\partial}{\partial w_{k_j}^k} \\
&\quad + \ell^{-1} \sum_{k=1}^{\ell} \sum_{s=1}^{\ell} \sum_{k_i=1}^{n_k} \sum_{s_j=1}^{n_s} \frac{w_{k_i}^k w_{s_j}^s}{|w^k||w^s|} \frac{\partial \widetilde{F}}{\partial w_{k_i}^k} \frac{\partial}{\partial w_{s_j}^s}.
\end{aligned}
\tag{54}
$$

In the last two terms, $|w^k| = |w^j|$, so it suffices to compute $|w^1|$.

Let $\binom{k_j}{k}$ denote the component of $c_k^{k_j} = \frac{\partial}{\partial w_{k_j}^k}$ of $\nabla^{\mathcal{S}} \widetilde{F}_{\boldsymbol{w}}$. Label the first three terms on the last line of (54) by I, II, III. Then

1. In I, $\binom{k_j}{k} = \frac{\partial \widetilde{F}}{\partial w_{k_j}^k}$.

2. In II,

$$
\binom{k_j}{k} = \underbrace{-\frac{1}{|w^1|^2}}_{\text{independent of } k,k_j} \underbrace{\left( \sum_{k_i=1}^{n_k'} w_{k_i}^k \frac{\partial \widetilde{F}}{\partial w_{k_i}^k} \right)}_{\text{independent of } k_j} w_{k_j}^k.
$$

3. In III, by switching the $s$ and $k$ indices,

$$
\binom{k_j}{k} = \underbrace{\frac{\ell^{-1}}{|w^1|^2} \left( \sum_{s=1}^{\ell} \sum_{s_i=1}^{n_s'} w_{s_i}^s \frac{\partial \widetilde{F}}{\partial w_{s_i}^s} \right)}_{\text{independent of } k,k_j} w_{k_j}^k \qquad \square
$$

# G    Sobolev Natural Gradient

For $\mathcal{H} = H_s(\mathbb{R}^n, \mathbb{R}^m)$ a Sobolev space (detailed in Appendix B.2), as shown in (14), in order to compute the Sobolev Natural Gradient on $\mathcal{W}$, we have to efficiently compute the inverse of the Sobolev metric tensor $\widetilde{g}_{\mathcal{H},ij}$, which is analogous to the Fisher information matrix in Amari's natural gradient. In §G.1, we discuss an RKHS-based approximation of $\widetilde{g}_{\mathcal{H},ij}$. In §G.2, we provide a practical computational method of $\widetilde{g}_{\mathcal{H},ij}$ based on the Kronecker-factored approximation (Martens & Grosse, 2015; Grosse & Martens, 2016). In §G.4, we report experimental results on supervised learning benchmarks.

## G.1    An approximate computation of $\widetilde{g}_{\mathcal{H},ij}$

In general, it is not possible to exactly compute $\widetilde{g}_{\mathcal{H},ij} = \left\langle \frac{\partial \phi}{\partial w^i}, \frac{\partial \phi}{\partial w^j} \right\rangle_{\mathcal{H}}$ for $\mathcal{H} = H_s$. We therefore resort to approximation. Following (13), if we do a gradient descent (Dieuleveut et al., 2016) in the RKHS $\mathcal{H}$ with a

sequence of data points $(x_1, y_1), (x_2, y_2), \cdots, (x_T, y_T)$, under the initial condition $f_0(x) = 0$, we get

$$f_T(x) = -\sum_{t=1}^{T} \eta_t K_{\mathcal{H}}(x_t, x) \nabla L_z(f(x_t), y_t). \tag{55}$$

Therefore, the gradient descent iterate $f_T$ lies in the subspace $K_{\mathcal{X}}$ of $\mathcal{H}$ spanned by $\{K_{\mathcal{H}}(x_t, \cdot)\}_{t=1}^{T}$ for a given dataset $\mathcal{X} = \{x_t\}_{t=1}^{T}$. From the proof of Proposition 2.1 (in Appendix C), for any $f \in \mathcal{H}$, the projection is $P_{K_{\mathcal{X}}} f = K_{ij}^{-1} \langle f, K(x_i, \cdot) \rangle_{\mathcal{H}} K(x_j, \cdot)$, where $K_{ij} = \langle K(x_i, \cdot), K(x_j, \cdot) \rangle_{\mathcal{H}}$. We then approximate $\widetilde{g}_{\mathcal{H}, ij}$ as follows,

$$\begin{aligned}
\widetilde{g}_{\mathcal{H}, ij} &= \left\langle \frac{\partial \phi}{\partial w^i}, \frac{\partial \phi}{\partial w^j} \right\rangle_{\mathcal{H}} \approx \left\langle P_{K_{\mathcal{X}}} \frac{\partial \phi}{\partial w^i}, P_{K_{\mathcal{X}}} \frac{\partial \phi}{\partial w^j} \right\rangle_{\mathcal{H}} \\
&= \left\langle K_{ab}^{-1} \left\langle \frac{\partial \phi}{\partial w^i}, K(x_a, \cdot) \right\rangle_{\mathcal{H}} K(x_b, \cdot), K_{cd}^{-1} \left\langle \frac{\partial \phi}{\partial w^j}, K(x_c, \cdot) \right\rangle_{\mathcal{H}} K(x_d, \cdot) \right\rangle_{\mathcal{H}} \\
&= \left\langle \frac{\partial \phi}{\partial w^i}, K(x_a, \cdot) \right\rangle_{\mathcal{H}} K_{ac}^{-1} \left\langle \frac{\partial \phi}{\partial w^j}, K(x_c, \cdot) \right\rangle_{\mathcal{H}} \\
&= \frac{\partial \phi}{\partial w^i}(x_a) K_{ac}^{-1} \frac{\partial \phi}{\partial w^j}(x_c). \tag{56}
\end{aligned}$$

Note that by setting $K_{ij}$ to the identity matrix, (56) becomess the Gauss-Newton approximation of the natural gradient metric (Zhang et al., 2019c). For a practical implementation of (56) used in the next subsection, we only need to invert a $K_{ab}$ matrix of size $B \times B$, where $B$ is the mini-batch size of supervised training, which is quite manageable in practice.

Now all we need is to compute $K_{ij} = \langle K(x_i, \cdot), K(x_j, \cdot) \rangle_{\mathcal{H}}$ for all $x_i, x_j \in \mathcal{X}$. When $\mathcal{H}$ is the Sobolev space $H_s$, Rosenberg (2023) proves the following Lemma that computes $K_{ij} = d_{x_i}(x_j)$,

**Lemma G.1.** *For* $s = \dim(M) + 3$,

$$d_{x_i}(x) = C_n e^{-\|x - x_i\|}(1 + \|x - x_i\|), \tag{57}$$

*where $C_n$ is some constant only depends on $n$.*

Even with (56), exact computation of the pullback metric for neural networks is in general extremely hard. To efficiently approximate (56) in practice, in the next subsection, we adapt the Kronecker-factored approximation techniques (Martens & Grosse, 2015; Grosse & Martens, 2016).

## G.2 Kronecker-factored approximation of $\widetilde{g}_{\mathcal{H}}$

Kronecker-factored Approximation Curvature (K-FAC) has been successfully used to approximate the Fisher information matrix for natural gradient/Newton methods. We now use fully-connected layers as an example to show how to adapt K-FAC (Martens & Grosse, 2015) to approximate $\widetilde{g}_{\mathcal{H}}$ in (56). Approximation techniques for convolutional layers can be similarly adapted from (Grosse & Martens, 2016). We omit standard K-FAC derivations and only focus on critical steps that are adapted for our approximation purposes. For full details of K-FAC, see (Martens & Grosse, 2015; Grosse & Martens, 2016).

As with the K-FAC approximation to the Fisher matrix, we first assume that entries of $\widetilde{g}_{\mathcal{H}}$ corresponding to different network layers are zero, which makes $\widetilde{g}_{\mathcal{H}}$ a block diagonal matrix, with each block corresponding to one layer of the network.

In the notation of (Martens & Grosse, 2015), the $\ell$-th fully-connected layer is defined by

$$\boldsymbol{s}_\ell = \bar{W}_\ell \bar{\boldsymbol{a}}_{\ell-1} \quad \bar{\boldsymbol{a}}_\ell = \psi_\ell(\boldsymbol{s}_\ell),$$

where $\bar{W}_\ell = (W_\ell \ \boldsymbol{b}_\ell)$ denotes the matrix of layer bias and weights, $\bar{\boldsymbol{a}}_\ell = (\boldsymbol{a}_\ell^T \ 1)^T$ denotes the activations with an appended homogeneous dimension, and $\psi_\ell$ denotes the nonlinear activation function.

For $\widetilde{g}_{\mathcal{H}}^{(\ell)}$ the block of $\widetilde{g}_{\mathcal{H}}$ corresponding to the $\ell$-th layer, the argument of (Martens & Grosse, 2015) applied to (56) gives

$$\widetilde{g}_{\mathcal{H}}^{(\ell)} = \mathbb{E}_K \left[ \bar{\boldsymbol{a}}_{\ell-1} \bar{\boldsymbol{a}}_{\ell-1}^T \otimes \mathcal{D}\boldsymbol{s}_\ell \mathcal{D}\boldsymbol{s}_\ell^T \right], \tag{58}$$

where $\mathcal{D}\boldsymbol{s}_\ell = \frac{\partial \phi}{\partial \boldsymbol{s}_\ell}$, $\otimes$ denotes the Kronecker product, and $\mathbb{E}_K$ is defined by

$$\mathbb{E}_K \left[ X \otimes Y \right] = X(x_i) K_{ij}^{-1} Y(x_j).$$

Just as K-FAC pushes the expectation of $\mathbb{E}\left[ \bar{\boldsymbol{a}}_{\ell-1} \bar{\boldsymbol{a}}_{\ell-1}^T \otimes \mathcal{D}\boldsymbol{s}_\ell \mathcal{D}\boldsymbol{s}_\ell^T \right]$ inwards by assuming the independence of $\bar{\boldsymbol{a}}_{\ell-1}$ and $\boldsymbol{s}_\ell$, we apply the same trick to (58) by assuming the following $K^{-1}$-independence between $\bar{\boldsymbol{a}}_{\ell-1}$ and $\boldsymbol{s}_\ell$:

$$\mathbb{E}_K \left[ \bar{\boldsymbol{a}}_{\ell-1} \bar{\boldsymbol{a}}_{\ell-1}^T \otimes \mathcal{D}\boldsymbol{s}_\ell \mathcal{D}\boldsymbol{s}_\ell^T \right] = \mathbb{E}_K \left[ \bar{\boldsymbol{a}}_{\ell-1} \bar{\boldsymbol{a}}_{\ell-1}^T \right] \otimes \mathbb{E}_K \left[ \mathcal{D}\boldsymbol{s}_\ell \mathcal{D}\boldsymbol{s}_\ell^T \right]. \tag{59}$$

The rest of the computation follows the standard K-FAC for natural gradient. Let $\boldsymbol{A}_{\ell-1}$ and $\boldsymbol{S}_\ell$ be the Kronecker factors

$$\boldsymbol{A}_{\ell-1} = \mathbb{E}_K \left[ \bar{\boldsymbol{a}}_{\ell-1} \bar{\boldsymbol{a}}_{\ell-1}^T \right] = \bar{\boldsymbol{a}}_{\ell-1}(x_i) K_{ij}^{-1} \bar{\boldsymbol{a}}_{\ell-1}^T(x_j), \tag{60}$$

$$\boldsymbol{S}_\ell = \mathbb{E}_K \left[ \mathcal{D}\boldsymbol{s}_\ell \mathcal{D}\boldsymbol{s}_\ell^T \right] = \mathcal{D}\boldsymbol{s}(x_i)_\ell K_{ij}^{-1} \mathcal{D}\boldsymbol{s}_\ell^T(x_j). \tag{61}$$

Then our natural gradient for the $\ell$-th layer can be computed efficiently by solving the linear system,

$$\left( \widetilde{g}_{\mathcal{H}}^{(\ell)} \right)^{-1} \text{vec}(V_\ell) = (\boldsymbol{A}_{\ell-1} \otimes \boldsymbol{S}_\ell)^{-1} \text{vec}(V_\ell) = \left( \boldsymbol{A}_{\ell-1}^{-1} \otimes \boldsymbol{S}_\ell^{-1} \right) \text{vec}(V_\ell) = \text{vec}(\boldsymbol{S}_\ell^{-1} V_\ell \boldsymbol{A}_{\ell-1}^{-1}),$$

where $\text{vec}(V_\ell)$ denotes the vector form of the Euclidean gradients of loss with respect to the parameters of the $\ell$-th layer. All Kronecker factors $\boldsymbol{A}_\ell$ and $\boldsymbol{S}_\ell$ are estimated by moving averages over training batches.

### G.3 Summary of Sobolev Natural Gradient Algorithm

Lemma §4.1 provides the following basic formula for our proposed Sobolev Natural Gradient,

$$P \nabla \ell_{L,f} = \widetilde{g}_{\mathcal{H}}^{k\ell} \sum_i \left( \nabla^E L_z(f(x_i), y_i) \cdot_E \left. \frac{\partial \phi}{\partial w^k} \right|_{x_i} \right) \frac{\partial \phi}{\partial w^\ell}, \tag{62}$$

where $L = L(z, y) \colon \mathbb{R}^n \times \mathbb{R}^n \to \mathbb{R}$ is some loss function and $\nabla^E L_z$ denotes the standard Euclidean gradient of $L$ w.r.t. its first input $z$. Note that summands on the RHS of (62) are just standard Euclidean gradient of $L$ w.r.t. model parameters $w$'s evaluated on training pairs $(x_i, y_i)$, which can be obtained by standard back-propagation. Then the only extra computation needed is the term $\widetilde{g}_{\mathcal{H}}^{k\ell}$, which is the inverse of the Sobolev metric tensor $\widetilde{g}_{\mathcal{H},ij}$. Applying this extra inverse of metric tensor is the main computational difference of all natural gradient methods vs. standard Euclidean gradients. For instance, Amari's original natural gradient applies the inverse of the Fisher information matrix.

The approximate computation technique for $\widetilde{g}_{\mathcal{H},ij}$ is addressed in §G.1, given by the following formula,

$$\widetilde{g}_{\mathcal{H},ij} = \frac{\partial \phi}{\partial w^i}(x_a) K_{ac}^{-1} \frac{\partial \phi}{\partial w^j}(x_c), \tag{63}$$

where $K_{ac} = d_{x_a}(x_c)$ can be computed by Lemma G.1 as following,

$$K_{ac} = C_n e^{-\|x_a - x_c\|} (1 + \|x_a - x_c\|). \tag{64}$$

As discussed in §G.1 and detailed in §G.2, in practice, (64) is the only extra computation step of our approach compared to the baseline. In fact, if $K_{ac}$ is set to the identity matrix, (63) becomes exactly the Gauss-Newton approximation of Amari's natural gradient metric (Zhang et al., 2019c), which is the baseline of all our experiments. This is another reason that we use the PyTorch codebase (Wang, 2019) of (Zhang et al., 2019c) to implement our algorithm and follow the experimental setup of (Zhang et al., 2019c) in our experiments.

Following the notations of §G.2, in Algorithm 2 below, we first copy (with slight tweaking for readability) the Algorithm 1 of (Zhang et al., 2019c), which gives the common algorithmic steps of our approach and the baseline, where we highlight in red the key step that our approach differs underneath from the baseline. We then zoom in on the highlighted step and provide computational details of it in Algorithm 3.

---

**Algorithm 2** K-FAC with weight decay (Zhang et al., 2019c)

---

**Require**: $\eta$ : stepsize
**Require**: $\beta$ : weight decay
**Require**: stats and inverse update intervals $T_{stats}$ and $T_{inv}$
**Init**: $\{\bar{W}_\ell\}_{\ell=1}^L$, $\{S_\ell\}_{\ell=1}^L$, $\{A_{\ell-1}\}_{\ell=1}^L$, $k = 0$
  **while** stopping criterion not met **do**
      $k \leftarrow k + 1$
      **if** $k \mod 0 \pmod{T_{stats}}$ **then**
         Update the factors $\{S_\ell\}_{\ell=1}^L$, $\{A_{\ell-1}\}_{\ell=1}^L$ with moving average
      **end if**
      **if** $k \mod 0 \pmod{T_{inv}}$ **then**
         Calculate the inverses $\{[S_\ell]^{-1}\}_{\ell=1}^L$, $\{[A_{\ell-1}]^{-1}\}_{\ell=1}^L$
      **end if**
      $V_\ell = \nabla_{\bar{W}_\ell} L$
      $\bar{W}_\ell \leftarrow \bar{W}_\ell - (\eta[A_{\ell-1}]^{-1}V_\ell[S_\ell]^{-1} + \beta \cdot \bar{W}_\ell)$
  **end while**

---

---

**Algorithm 3** Update of factors $\{S_\ell\}_{\ell=1}^L$ and $\{A_{\ell-1}\}_{\ell=1}^L$

---

**Compute**: $K_{ij}$ by (64) for all samples of the current training batch
**Compute**: the inverse $K_{ij}^{-1}$
**Update**: $A_{\ell-1}$ by (60)
**Update**: $S_\ell$ by (61)

---

### G.4 Experimental results

Given that comparisons between natural gradient descent and Euclidean gradient descent have been conducted extensively in the literature (Zhang et al., 2019c;a) and that our proposed approach ends up being a new variant of natural gradient, our numerical experiments focus on comparing our Sobolev natural gradient (ours) with the most widely-used natural gradient (baseline) originally proposed by (Amari, 1998) in the supervised learning setting.

Since we borrow the K-FAC approximation techniques in designing an efficient computational method for our Sobolev natural gradient, we follow the settings of (Zhang et al., 2019c), which applies the K-FAC approximation to Amari's natural gradient, to test both gradient methods on the VGG16 neural network on the CIFAR-10 and the CIFAR-100 image classification benchmarks. Our implementations and testbed are based on the PyTorch K-FAC codebase (Wang, 2019) provided by one of the co-authors of (Zhang et al., 2019c). We did a grid search for the optimal combination of the hyper-parameters for the baseline: learning rate, weight decay factor, and the damping factor. We end up using learning rate 0.01, weight decay 0.003, and damping 0.03 for the baseline through out our experiments reported in Figure 3 and Figure 4. For learning rate scheduling of the baseline, we follow the suggested default scheduling of (Wang, 2019) to decrease the learning rate to 1/10 after every 40% of the total training epochs.

For our method, we use the same learning rate 0.01, weight decay 0.003, and damping 0.03 as the baseline throughout our experiments. We only tune two extra hyper-parameters for our method,

- For the learning rate scheduling, we decrease our learning rate to 1/5 after the first 40% of total epochs, and decrease again to 1/5 after another 20% of total epochs.

- From (57), the Sobolev kernel $K_{ij}(d_{x_i}(x_j))$ contains an exponential function with the exponent $-\|x_i - x_j\|$, as a result, this kernel reduces to identity matrix if the set of points $\{x_i\}$ are not close to each other. Given that input data points are by default normalized when loaded in from these classification benchmarks, we just introduce for our method a scaling factor to further down scale all input data. We fix this scaling factor to be 20 throughout our experiments.

We also run the baseline under different combinations of these tuned extra hyper-parameters, as shown in Figure 5 and Figure 6, they cannot bring any benefit to the baseline. All experiments are run on an desktop with an Intel i9-7960X 16-core CPU, 64GB memory, and an a GeForce RTX 2080Ti GPU.

When comparing the natural gradient with the SGD, (Zhang et al., 2019c) trains natural gradients for 100 epochs while training the SGD for 200 epochs, highlighting the training efficiency of natural gradient methods. Following the same philosophy, in comparing the two natural gradient variants, we further shorten the training epochs to 50 in all our experiments, and we have found that the final performance is almost on par with that trained with 100 epochs. The training and testing behavior of ours vs. the baseline on CIFAR-10 and CIFAR-100 is shown in Figure 3 and Figure 4, respectively. While the final testing performance of both natural gradient variants are similar, as expected, our method shows a clear advantage regarding convergence speed, with the margin increased on the more challenging CIFAR-100 benchmark.

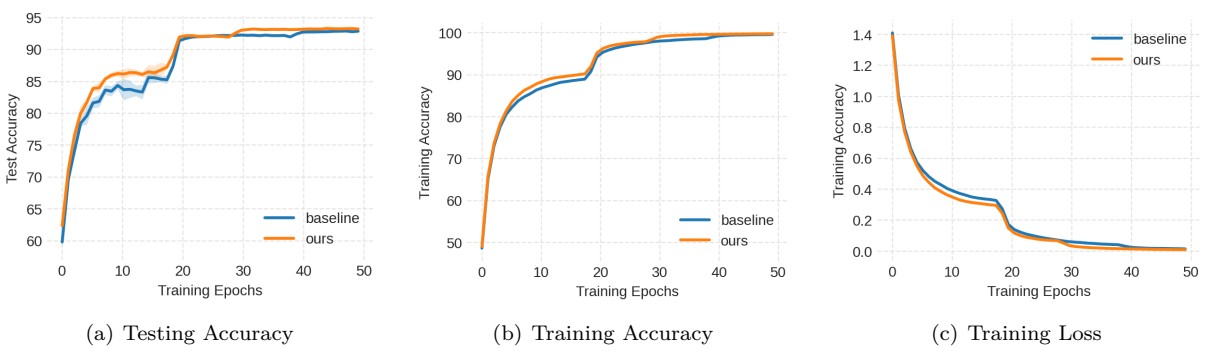

(a) Testing Accuracy  (b) Training Accuracy  (c) Training Loss

Figure 3: Training and testing behaviors of Sobolev Natural Gradient (ours) vs Amari's Natural Gradient (baseline) on CIFAR-10. Results are averaged over four runs of different random seeds, with the shaded area corresponding to the standard deviation.

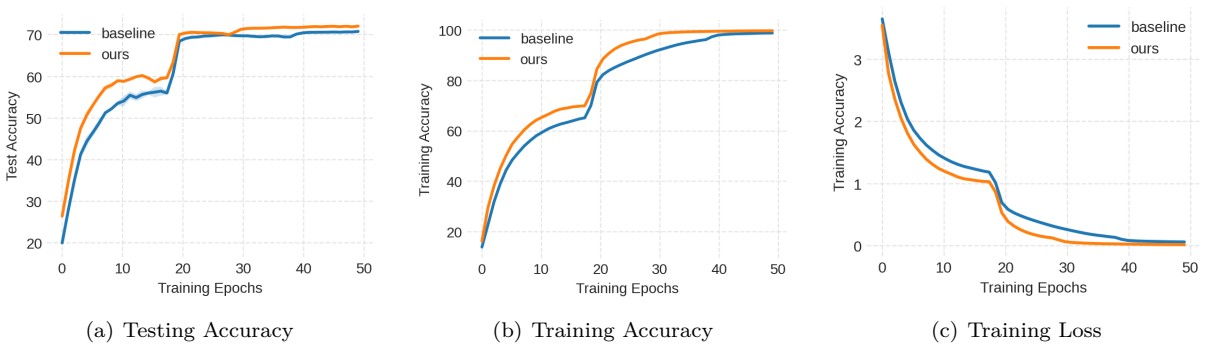

(a) Testing Accuracy  (b) Training Accuracy  (c) Training Loss

Figure 4: Training and testing behaviors of Sobolev Natural Gradient (ours) vs Amari's Natural Gradient (baseline) on CIFAR-100. Results are averaged over four runs of different random seeds, with the shaded area corresponding to the standard deviation.

## H  Pullback metric vs. pushforward metric: an example involving flatness

Our GTK theory specifies a Riemannian metric on the function space which makes the standard gradient, at least locally, as "natural" as the natural gradient in capturing the intrinsic structure of the parameterized function space. In this section, we compare the pullback metric and the pushforward metric from a purely mathematical viewpoint, emphasizing that the pullback metric has better compatibility properties. For example, suppose $\mathcal{W} \subset \mathcal{W}'$, and $\phi \colon \mathcal{W} \to \mathcal{M}$ extends to $\phi' \colon \mathcal{W}' \to \mathcal{M}$. Then the pullback metrics behave

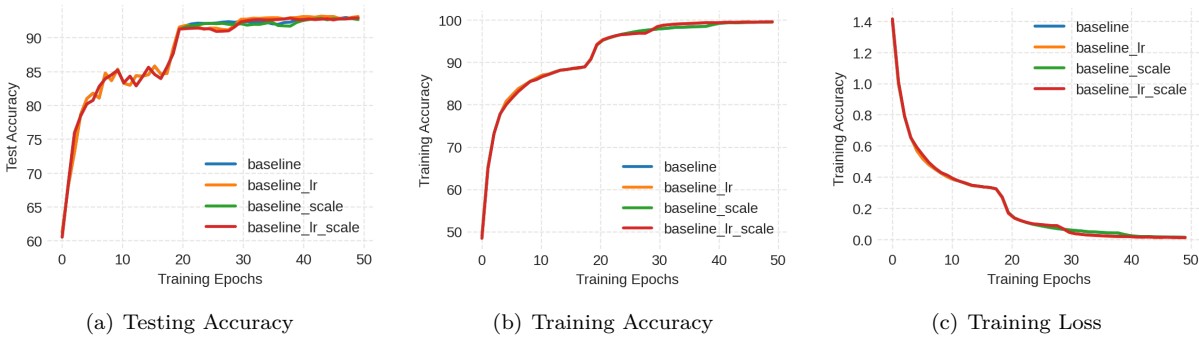

(a) Testing Accuracy        (b) Training Accuracy        (c) Training Loss

Figure 5: Training and testing behaviors of Amari's Natural Gradient (baseline) on CIFAR-10. *baseline_lr* is the baseline method with the same learning rate scheduling as our method. *baseline_scale* is the baseline method with the same input scaling as our method. *baseline_lr_scale* is the baseline method with the same learning rate scheduling and input scaling as our method.

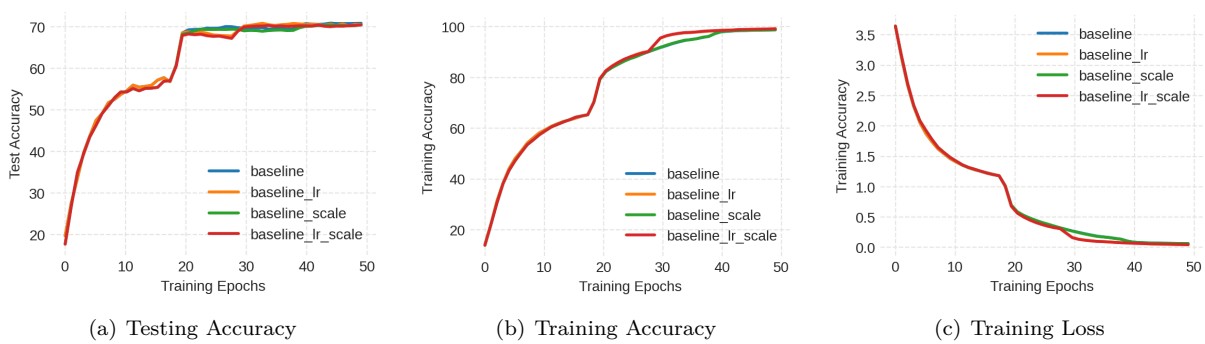

(a) Testing Accuracy        (b) Training Accuracy        (c) Training Loss

Figure 6: Training and testing behaviors Amari's Natural Gradient (baseline) on CIFAR-100. *baseline_lr* is the baseline method with the same learning rate scheduling as our method. *baseline_scale* is the baseline method with the same input scaling as our method. *baseline_lr_scale* is the baseline method with the same learning rate scheduling and input scaling as our method.

well: $(\phi')^*\bar{g} = \phi^*\bar{g}$ for a metric $\bar{g}$ on $\mathcal{M}$. In contrast, there is no relationship in general between $(\phi')_*g_E$ and $\phi_*g_E$.

A key point is that *pullback metrics are independent of change of coordinates on the parameter space $\mathcal{W}$, while pushforward metrics depend on a choice of coordinates.* As a simple example, if we scale the standard coordinates $(x^1, \ldots, x^k)$ on $\mathcal{W} \subset \mathbb{R}^k$ to $(y^1, \ldots, y^k) = (2x^1, \ldots, 2x^k)$, then in (1) the pushforward metric $\phi_*g_E = \sum_i(\partial\phi/\partial x^i)\otimes(\partial\phi/\partial x^i)$ of the Euclidean metric on $\mathcal{W}$ changes to $\sum_i(\partial\phi/\partial y^i)\otimes(\partial\phi/\partial y^i) = (1/4)\phi_*g_E$. More generally, if $y = \alpha(x)$ is a change of coordinates on $\mathcal{W}$ given by a diffeomorphism $\alpha$, the pushforward metrics for the two coordinate systems will be very different.

In contrast, the pullback metric is well behaved ("natural" in math terminology) with respect to coordinate changes. If $\bar{g}$ is a Riemannian metric on $\mathcal{M}$, then $(\phi \circ \alpha)^*\bar{g} = \alpha^*(\phi^*\bar{g})$. In particular, for $v, w$ tangent vectors at a point in $\mathcal{W}$, we have

$$\langle v, w \rangle_{(\phi \circ \alpha)^*\bar{g}} = \langle d\alpha(v), d\alpha(w) \rangle_{\phi^*\bar{g}}. \tag{65}$$

Since $v$ in the $x$-coordinate chart is identified with $d\alpha(v)$ in the $y = \alpha(x)$-coordinate chart, (65) says that the inner product of tangent vectors to $\mathcal{W}$ is independent of chart for pullback metrics.

For the rest of this section, we give a concrete example in machine learning where these properties of the pullback metric have an explicit advantage over the pushforward metric. In particular, we consider the various notions of flatness associated to the basic setup (1) near a minimum. In (Dinh et al., 2017, §5),

$\epsilon$-flatness of $\widetilde{F} \colon \mathcal{W} \to \mathbb{R}$ near a minimum $w_0$ is defined by *e.g.,* measuring the Euclidean volume of maximal sets $S \subset \mathcal{W}$ on which $F(w) < F(w_0) + \epsilon$ for all $w$ in $S$. The authors note that their definitions of flatness are not independent of scaling the Euclidean metric by different factors in different directions. Here the pushforward of the Euclidean metric is implicitly used. As above, scaling is the simplest version of a change of coordinates of the parameter space. Since this definition of flatness is not coordinate free, it has no intrinsic differential geometric meaning.

This issue is discussed in (Dinh et al., 2017), Fig. 5. Under simple scaling of the $x$-axis, the graph of a function $f \colon \mathbb{R} \to \mathbb{R}$ will "look different" Of course, $f$ is independent of scaling, but the graph looks different to an eye with an implicit fixed scale for the $x$-axis, *i.e.*, from the point of view of a fixed coordinate chart.

We instead obtain the following coordinate free notion of flatness by measuring volumes in the pullback metric $\phi^* \bar{g}$.

**Definition H.1.** *In the notation of (1), the $\epsilon$-flatness of $\widetilde{F}$ near a local minimum $w_0$ is $\int_S \mathrm{dvol}_{\phi^* \bar{g}}$ where $S$ is a maximal connected set such that $\widetilde{F}(S) \subset (\widetilde{F}(w_0), \widetilde{F}(w_0) + \epsilon)$.*

Since $\int_S \mathrm{dvol}_{(\phi \circ \alpha)^* \bar{g}} = \int_S \mathrm{dvol}_{\phi^* \bar{g}} = \int_{\phi(S)} \mathrm{dvol}_{\bar{g}}$, for any orientation preserving diffeomorphism of $\mathcal{W}$, $\epsilon$-flatness is independent of coordinates on $\mathcal{W}$, and can be measured on $S$ or on $\mathcal{M}$. In contrast, the Euclidean volume of $S$ is not related to the volume of $\phi(S) \subset \mathcal{M}$.

# I MLP with ReLU Activation - General Case with Bias Vectors

In Appendix E, we discuss the gradient for an MLP and with no bias vectors. In this Appendix, we give the necessary changes for an MLP with bias vectors.

The setup is an MLP with $\ell$ layers with ReLU activation function, with parameters $w_r^k$ in the $k$-th layer, for $r = 1, \ldots, n_k$, where $n_k$ is the number of matrix entries plus the number of components of the bias vector in the $k$-th layer. The space of parameter vectors is $\mathcal{W} = \{\boldsymbol{w} = (w_1^1, \ldots, w_{n_\ell}^\ell)\}$, where each parameter in $\mathbb{R}$. Thus $\dim(\mathcal{W}) = \sum_\ell n_\ell$. The map $\phi \colon \mathcal{W} \to \mathrm{Maps}(\mathbb{R}^n, \mathbb{R}^M)$ takes a parameter to the associated MLP.

We abbreviate the subvector for the $k$-th level by $w^k = (w_1^k, \ldots, w_{n_k}^k)$, so $\boldsymbol{w} = (w^1, \ldots, w^\ell)$. We further refine $w^k = (w_1^k, \ldots, w_{a_k}^k, \bar{w}_1^k, \ldots \bar{w}_{c_k}^k) := (w_{A_k}, w_{b_k})$, where the matrix $A_k$ in the $k^{\mathrm{th}}$ layer has $a_k$ entries and the bias vector $b_k$ has $c_k$ entries. We may have $c_k = 0$ for some $k$. We can assume that $w^k \neq \boldsymbol{0}$ for all $k$, since a MLP with some $w^k = \boldsymbol{0}$ is trivial.

Define a multiplicative group $G = \{\boldsymbol{g} = (g_1, \ldots, g_\ell) : g_i > 0, \prod_{i=1}^\ell g_i = 1\}$. The map $G \to \mathbb{R}^\ell : (g_1, \ldots, g_\ell) \mapsto \sum_i \log g_i$ is a bijection to the plane $\sum_i x^i = 0$, so $G$ is a manifold of dimension $\ell - 1$. Then $G$ acts on $\mathcal{W}$ by

$$
\begin{aligned}
&(g_1, \ldots, g_\ell) \cdot (w^1, \ldots, w^\ell) \\
&= (\overbrace{(g_1 \cdot (w_1^1, \ldots, w_{a_1}^1), g_1 \cdot (\bar{w}_1^1, \ldots, \bar{w}_{c_1}^1))}^{(g_1 \cdot w_{A_1}, \, g_1 \cdot w_{b_1})}), \\
&\quad\ (g_2 \cdot (w_1^2, \ldots, w_{a_2}^2), g_1 g_2 \cdot (\bar{w}_1^2, \ldots, \bar{w}_{c_2}^2)), \\
&\qquad\qquad \vdots \\
&\quad\ (g_{\ell-1} \cdot (w_1^{\ell-1}, \ldots, w_{a_{\ell-1}}^{\ell-1}), g_1 g_2 \cdot \ldots \cdot g_{\ell-1} \cdot (\bar{w}_1^{\ell-1}, \ldots, \bar{w}_{c_{\ell-1}}^{\ell-1})), \\
&\quad\ \underbrace{(g_\ell \cdot (w_1^\ell, \ldots, w_{a_\ell}^\ell), (\bar{w}_1^\ell, \ldots, \bar{w}_{c_\ell}^\ell))}_{(g_\ell \cdot w_{A_\ell}, \, w_{b_\ell})}).
\end{aligned}
\tag{66}
$$

By Dinh et al. (2017, Appendix B),

$$
\phi(w^1, \ldots, w^\ell) = \phi(\boldsymbol{g} \cdot (w^1, \ldots, w^\ell)),
\tag{67}
$$

for all $\boldsymbol{g} \in G$. Thus, as in §5.1, $\phi$ is far from injective, and in fact is not an immersion.

**Proposition I.1.** *The dimension of $ker(d\phi)_{\boldsymbol{w}}$ at each $\boldsymbol{w} \in \mathcal{W}$ is at least $\ell - 1$. Thus $\phi$ is not an immersion. In fact, the span of*

$$\left\{ Z_r|_{\boldsymbol{w}} = \sum_{k=1}^{n_1} w_1^{k_1} \frac{\partial}{\partial w_1^{k_1}}\bigg|_{\boldsymbol{w}} + \sum_{k_2=1}^{c_2} \bar{w}_2^{k_2} \frac{\partial}{\partial \bar{w}_2^{k_2}}\bigg|_{\boldsymbol{w}} + \ldots + \sum_{k_{r-1}=1}^{c_{r-1}} \bar{w}_{r-1}^{k_{r-1}} \frac{\partial}{\partial \bar{w}_{r-1}^{k_{r-1}}}\bigg|_{\boldsymbol{w}} - \sum_{k_r=1}^{c_r} \bar{w}_r^{k_r} \frac{\partial}{\partial \bar{w}_r^{k_r}}\bigg|_{\boldsymbol{w}} \right\} \tag{68}$$

*is in $ker(d\phi)_{\boldsymbol{w}}$.*

*Proof.* For fixed $r \in \{2, \ldots, \ell\}$, take $g_1 > 0$, $g_r = g_1^{-1}$, and all other $g_i = 1$, and then take $(d/dg_1)|_{g_1=1}$ of both sides of (25). This gives

$$0 = \sum_{k=1}^{n_1} \frac{\partial \phi}{\partial w_1^{k_1}} w_1^{k_1} + \sum_{k_2=1}^{c_2} \frac{\partial \phi}{\partial \bar{w}_2^{k_2}} \bar{w}_2^{k_2} + \ldots + \sum_{k_{r-1}=1}^{c_{r-1}} \frac{\partial \phi}{\partial \bar{w}_{r-1}^{k_{r-1}}} \bar{w}_{r-1}^{k_{r-1}} - \sum_{k_r=1}^{c_r} \frac{\partial \phi}{\partial \bar{w}_r^{k_r}} \bar{w}_r^{k_r}, \tag{69}$$

where the first sum is over all the neurons in the first layer, and the other sums are over the bias vector neurons. (69) is equivalent to

$$d\phi \left( \sum_{k=1}^{n_1} w_1^{k_1} \frac{\partial}{\partial w_1^{k_1}} + \sum_{k_2=1}^{c_2} \bar{w}_2^{k_2} \frac{\partial}{\partial \bar{w}_2^{k_2}} + \ldots + \sum_{k_{r-1}=1}^{c_{r-1}} \bar{w}_{r-1}^{k_{r-1}} \frac{\partial}{\partial \bar{w}_{r-1}^{k_{r-1}}} - \sum_{k_r=1}^{c_r} \bar{w}_r^{k_r} \frac{\partial}{\partial \bar{w}_r^{k_r}} \right) = 0. \tag{70}$$

Thus the $\ell - 1$ vectors in $T_{\boldsymbol{w}}\mathcal{W}$

$$Z_r|_{\boldsymbol{w}} = \sum_{k=1}^{n_1} w_1^{k_1} \frac{\partial}{\partial w_1^{k_1}}\bigg|_{\boldsymbol{w}} + \sum_{k_2=1}^{c_2} \bar{w}_2^{k_2} \frac{\partial}{\partial \bar{w}_2^{k_2}}\bigg|_{\boldsymbol{w}} + \ldots + \sum_{k_{r-1}=1}^{c_{r-1}} \bar{w}_{r-1}^{k_{r-1}} \frac{\partial}{\partial \bar{w}_{r-1}^{k_{r-1}}}\bigg|_{\boldsymbol{w}} - \sum_{k_r=1}^{c_r} \bar{w}_r^{k_r} \frac{\partial}{\partial \bar{w}_r^{k_r}}\bigg|_{\boldsymbol{w}} \tag{71}$$

are in $ker(d\phi)_{\boldsymbol{w}}$. Since $w_j \neq 0$ for all $j$, these vectors are nonzero. They are clearly linearly independent. Thus the span of the $Z_r$ is contained in $ker(d\phi)_{\boldsymbol{w}^0}$. $\square$

As in Appendix F, the $Z_r$ at $\boldsymbol{w}$ form a basis of $T_{\boldsymbol{w}}\mathcal{O}$ by a dimension count. Since $\phi$ is not an immersion for ReLU MLPs, as in §3.2 the natural gradient does not exist, and we must find a slice for the group action.

**Theorem I.1.** *Write $\mathcal{W} = \mathbb{R}^{n_1} \times \ldots \times \mathbb{R}^{n_\ell}$. A slice is given by*

$$\mathcal{S} = \{\lambda(v_1, \ldots, v_\ell) : (v_1, \ldots, v_\ell) \in S^{n_1-1} \times S^{n_2-2} \times \ldots \times S^{n_{\ell-1}-1} \times S^{a_\ell}, \lambda > 0\} \times \mathbb{R}^{c_\ell,*},$$

*where $S^{k-1}$ is the unit sphere in $\mathbb{R}^k$ and $\mathbb{R}^{k,*} = \mathbb{R}^k \setminus \{\boldsymbol{0}\}$.*

The orbit $\mathcal{O}_{\boldsymbol{w}} = \{\boldsymbol{g} \cdot \boldsymbol{w} : \boldsymbol{g} \in G\}$ is contained in (and probably equals) a level set of $\phi$. It is unlikely that any nonzero vector in $T_{\boldsymbol{w}}\mathcal{S}$ lies in $ker(d\phi)_{\boldsymbol{w}}$. Since $\dim(T_{\boldsymbol{w}}\mathcal{S}) + \dim(\text{Span}\{Z_r\}) = \dim(\mathcal{W})$, we expect that $ker(d\phi)_{\boldsymbol{w}} = \text{Span}\{Z_r\}$ in Prop. I.1, so that $\phi|_{\mathcal{S}}$ is an immersion.

*Proof.* We prove that every element $\boldsymbol{w} = (w_1, \ldots, w_\ell) \in \mathcal{W}$ lies on the orbit through some slice element. Write $w_k = (A_k, b_k)$ with $A_k = (w_{k,1}, \ldots, w_{k,a_k})$, $b_k = (w_{a_k+1}, \ldots, w_{n_k})$, where $a_k$ is the number of entries of the matrix in the $k^{\text{th}}$ layer. Set

$$g_1' = |(A_1, b_1)|^{-1}, \ g_2' = |(A_2, g_1'b_2)|^{-1}, \ldots, \ g_{\ell-1}' = |(A_{\ell-1}, \ g_{\ell-2}'g_{\ell-3}' \cdot \ldots \cdot g_1'b_{\ell-1})|^{-1}, \ g_\ell' = |A_\ell|^{-1}. \tag{72}$$

Then

$$(g_1', \ldots, g_\ell') \cdot ((A_1, b_1), \ldots, (A_\ell, b_\ell))$$
$$= ((g_1'A_1, g_1'b_1), (g_2'A_2, g_2'g_1'b_2), \ldots, (g_{\ell-1}'A_{\ell-1}, g_{\ell-1}' \cdot \ldots \cdot g_1'b_{\ell-1}), (g_\ell'A_\ell, b_\ell)) \tag{73}$$
$$\in S^{n_1} \times S^{n_2} \times \ldots \times S^{n_{\ell-1}} \times S^{a_\ell} \times \mathbb{R}^{c_\ell}.$$

Set

$$g_k = \frac{g_k'}{\left( \prod_{i=1}^{\ell} g_i' \right)^{1/\ell}}.$$

Then $\boldsymbol{g} = (g_1, \ldots, g_\ell) \in G$, and $\boldsymbol{g} \cdot \boldsymbol{w} \in \mathcal{S}$. $\square$

For an $\ell$-layer MLP with ReLU activation function, the slice $\mathcal{S}$ in Theorem I.1 has codimension $\ell - 1$ in $\mathcal{W}$, contains all the information in $\phi\colon \mathcal{W} \to \mathcal{M}(\mathbb{R}^n, \mathbb{R}^m)$, and probably has $\phi|_S$ an immersion.

§5.1 suggests three approaches to treat the non-immersion case. We take a basis $\{\boldsymbol{r} := c_0^0, c_k^{k_i} : k = 1, \ldots, \ell; k_i = 1, \ldots, n_k - 1\}$ of the tangent space $T_{\boldsymbol{w}}S$ for the slice $\mathcal{S}$ in Thm. I.1. We use spherical coordinates $(r_k, \psi_k^{k_i}), k_i = 1, \ldots, n_k - 1$, on the $k^{\text{th}}$ layer, for $k = 1, \ldots, \ell - 1$, while for the $\ell^{\text{th}}$ layer we use $(r_\ell, \psi_\ell^{\ell_i})$, $\ell_i = 1, \ldots, a_\ell$, and rectangular coordinates $(x^1, \ldots, x^{b_\ell})$ on $\mathbb{R}^{b_\ell,*}$. Then $\boldsymbol{r} := \sum_k \partial/\partial r_k; c_k^{k_i} = \partial/\partial \psi_k^{k_i}$, $k = 1, \ldots, \ell - 1; c_\ell^{\ell_i} = \partial/\partial \psi_\ell^{\ell_i}, \ell_i = 1, \ldots, a_\ell; c_\ell^{\ell_i} = \partial/\partial x^{\ell_i}, \ell_i = a_\ell + 1, \ldots, n_\ell$. The Euclidean metric/dot product restricts to a metric on $\mathcal{S}$ with metric tensor

$$h = h_{(k,k_i),(s,s_j)} = c_k^{k_i} \cdot c_s^{s_j} \tag{74}$$

$$= \begin{cases} 0, & \text{for } k \neq s, \\ 0, & \text{for } k = s \neq 0, k_i \neq s_j, \\ |w_k|^2 \sin^2(\psi_k^{n_k-1}) \cdot \sin^2(\psi_k^{n_k-2}) \cdot \ldots \cdot \sin^2(\psi_k^{k_i+1}), & \text{for } k = s \neq 0, \ell; k_i = s_j, \\ |w_\ell|^2 \sin^2(\psi_\ell^{n_\ell-1}) \cdot \sin^2(\psi_\ell^{n_\ell-2}) \cdot \ldots \cdot \sin^2(\psi_\ell^{k_i+1}), & \text{for } k = s = \ell, k_i = s_j \leq a_\ell, \\ 1, & \text{for } k = s = \ell, k_i = s_j > a_\ell, \\ \ell, & \text{for } k = s = 0, i = j = 0. \end{cases}$$

Here the metric is computed at $\boldsymbol{w} = (w_1, \ldots, w_\ell) \in \mathcal{S}$. Note that $|w_k|^2 = r_k^2$ independent of $k$ (except for $|w_{\ell,A}|^2 = r_\ell^2$). $h$ is diagonal, so $h^{-1}$ is easy to compute. (74) follows from (48).

We can use this metric in any of Approaches I-III of §5. The implementation of gradient descent in any approach is similar to §5, although moving a parameter vector back into the slice is a little more complicated, as we now explain.

To implement gradient descent on $\mathcal{W}$, start with $\boldsymbol{w}^0 \in \mathcal{S}$ and take $\bar{\boldsymbol{w}}^1 = \boldsymbol{w}^0 + \alpha \cdot d\phi^{-1}\left(\nabla^\Phi L_{\ell,\phi(\boldsymbol{w}^0)}\right)$, where $\alpha$ is the step size. Move $\bar{\boldsymbol{w}}^1$ along its $G$-orbit back to $\boldsymbol{w}^1 \in \mathcal{S}$ as follows: for $\bar{\boldsymbol{w}}^1 = ((A_1, b_1), \ldots, (A_\ell, b_\ell))$, define $g_i'$ by

$$g_1' = |(A_1, b_1)|^{-1}, \ g_2' = |(A_2, g_1' b_2)|^{-1}, \ldots, \ g_{\ell-1}' = |(A_{\ell-1}, \ g_{\ell-2}' g_{\ell-3}' \cdot \ldots \cdot g_1' b_{\ell-1})|^{-1}, \ g_\ell' = |A_\ell|^{-1},$$

and set

$$\boldsymbol{w}^1 = B\left((g_1' A_1, g_1' b_1), (g_2' A_2, g_2' g_1' b_2), \ldots, (g_{\ell-1}' A_{\ell-1}, g_{\ell-1}' \cdot \ldots \cdot g_1' b_{\ell-1}), (g_\ell' A_\ell, B^{-1} b_\ell)\right), \tag{75}$$

where $B = \left(\prod_{i=1}^\ell g_i'\right)^{-1/\ell}$. This ensures both that $(Bg_1', \ldots, Bg_\ell') \in G$ and that $b_\ell$ is not modified. Now continue with $\boldsymbol{w}^1$.

### I.1 Loss function gradient: Approach I

In particular, for Approach I we need:

**Proposition I.2.** *The gradient $\nabla^\mathcal{S} \widetilde{F}$, for $\widetilde{F}\colon \mathcal{W} \to \mathbb{R}$, is given by*

$$\nabla^\mathcal{S} \widetilde{F}_{\boldsymbol{w}} = \nabla^{Euc,\mathcal{W}} \widetilde{F}_{\boldsymbol{w}} - \frac{1}{|w^1|^2} \sum_{k=1}^\ell \left(\sum_{k_i=1}^{n_k'} w_{k_i}^k \frac{\partial \widetilde{F}}{\partial w_{k_i}^k}\right) w_{k_j}^k \frac{\partial}{\partial w_{k_j}^k} + \frac{\ell^{-1}}{|w_1|^2} \left(\sum_{s=1}^\ell \sum_{s_i=1}^{n_s'} w_{s_i}^s \frac{\partial \widetilde{F}}{\partial w_{s_i}^s}\right) w_{k_j}^k \frac{\partial}{\partial w_{k_j}^k},$$

*where*

$$n_k' = \begin{cases} n_k, & k \neq \ell, \\ a_k, & k = \ell. \end{cases}$$

*Proof.* For $\boldsymbol{w}$ in the slice $\mathcal{S}$, write $\boldsymbol{w} = (w_1, \ldots, w_\ell)$ with $w_k = (w_k^1, \ldots, w_k^{n_k})$. Since the gradient is independent of coordinates, on the $k^{\text{th}}$ layer, $k = 1, \ldots, \ell - 1$, for any $\widetilde{F} \colon \mathcal{W} \to \mathbb{R}$ and any $\boldsymbol{w} = (w_1, \ldots, w_k) \in \mathcal{W}$, we have

$$
\begin{aligned}
\nabla^{Euc,k} \widetilde{F}_{w^k} &= \sum_{k_i=1}^{n_k} \frac{\partial \widetilde{F}}{\partial w_{k_i}^k} \frac{\partial}{\partial w_{k_i}^k} \\
&= h_k^{0,0} \frac{\partial \widetilde{F}}{\partial r_k} \frac{\partial}{\partial r_k} + h^{0,k_i} \frac{\partial \widetilde{F}}{\partial r_k} \frac{\partial}{\partial \psi_k^{k_i}} + h^{k_i,0} \frac{\partial \widetilde{F}}{\partial \psi_k^{k_i}} \frac{\partial}{\partial r_k} + h^{k_i k_j} \frac{\partial \widetilde{F}}{\partial \psi_k^{k_i}} \frac{\partial}{\partial \psi_k^{k_j}} \\
&= \frac{\partial \widetilde{F}}{\partial r_k} \frac{\partial}{\partial r_k} + \sum_{k_i=1}^{n_k-1} \left( \sum_{b=1}^{k_i+1} (w_b^k)^2 \right)^{-1} \frac{\partial \widetilde{F}}{\partial \psi_k^{k_i}} \frac{\partial}{\partial \psi_k^{k_i}}.
\end{aligned}
$$

The right hand side is evaluated at $w_k$. As before, $h_k^{0,0} := \langle \partial_{r_k}, \partial_{r_k} \rangle^{-1} = 1$, we abbreviate $h^{(k,k_i),(k,k_j)}$ by $h^{k_i,k_j}$ when the context is clear, and we have used (52). For $k = \ell$, we use spherical coordinates $(r_\ell, \psi_\ell^k)$ on the first $a_\ell$ components and rectangular coordinates on the bias vector components. This gives

$$
\begin{aligned}
\nabla^{Euc,\ell} \widetilde{F}_{w^\ell} &= \sum_{\ell_i=1}^{n_\ell} \frac{\partial \widetilde{F}}{\partial w_{\ell_i}^\ell} \frac{\partial}{\partial w_{\ell_i}^\ell} \\
&= \frac{\partial \widetilde{F}}{\partial r_\ell} \frac{\partial}{\partial r_\ell} + \sum_{\ell_i=1}^{a_\ell-1} \left( \sum_{b=1}^{\ell_i+1} (w_b^k)^2 \right)^{-1} \frac{\partial \widetilde{F}}{\partial \psi_\ell^{\ell_i}} \frac{\partial}{\partial \psi_\ell^{\ell_i}} + \sum_{\ell_i=a_\ell+1}^{n_\ell} \frac{\partial \widetilde{F}}{\partial w_{\ell_i}^\ell} \frac{\partial}{\partial w_{\ell_i}^\ell}.
\end{aligned}
$$

To compress the notation, set

$$
n_k' = \begin{cases} n_k, & k \neq \ell, \\ a_k, & k = \ell. \end{cases}
$$

Using a mixture of ordinary summation and summation convention, we have

$$
\begin{aligned}
\nabla^{\mathcal{S}} \widetilde{F}_{\boldsymbol{w}} &= h^{0,0} \frac{\partial \widetilde{F}}{\partial \boldsymbol{r}} \frac{\partial}{\partial \boldsymbol{r}} + \sum_{k=1}^{\ell-1} h^{k_i k_j} \frac{\partial \widetilde{F}}{\partial \psi_k^{k_i}} \frac{\partial}{\partial \psi_k^{k_j}} + \sum_{\ell_i,\ell_j=1}^{a_\ell} h^{\ell_i \ell_j} \frac{\partial \widetilde{F}}{\partial \psi_\ell^{\ell_i}} \frac{\partial}{\partial \psi_\ell^{\ell_j}} + \sum_{\ell_i=a_\ell+1}^{n_\ell} \frac{\partial \widetilde{F}}{\partial w_{\ell_i}^\ell} \frac{\partial}{\partial w_{\ell_i}^\ell} \\
&= \ell^{-1} \left( \sum_{k=1}^{\ell} \frac{\partial \widetilde{F}}{\partial r_k} \right) \left( \sum_{s=1}^{\ell} \frac{\partial}{\partial r_s} \right) + \sum_{k=1}^{\ell} \sum_{k_i=1}^{n_k'-1} \left( \sum_{b=1}^{k_i+1} (w_b^k)^2 \right)^{-1} \frac{\partial \widetilde{F}}{\partial \psi_k^{k_i}} \frac{\partial}{\partial \psi_k^{k_i}} + \sum_{\ell_i=a_\ell+1}^{n_\ell} \frac{\partial \widetilde{F}}{\partial w_{\ell_i}^\ell} \frac{\partial}{\partial w_{\ell_i}^\ell} \\
&= \nabla^{Euc,\mathcal{W}} \widetilde{F} - \sum_{k=1}^{\ell} \frac{\partial \widetilde{F}}{\partial r_k} \frac{\partial}{\partial r_k} + \ell^{-1} \sum_{k=1}^{\ell} \sum_{s=1}^{\ell} \frac{\partial \widetilde{F}}{\partial r_k} \frac{\partial}{\partial r_s} \\
&= \nabla^{Euc,\mathcal{W}} \widetilde{F} - \sum_{k=1}^{\ell} \sum_{k_i,k_j=1}^{n_k'} \frac{w_{k_i}^k w_{k_j}^k}{|w^k|^2} \frac{\partial \widetilde{F}}{\partial w_{k_i}^k} \frac{\partial}{\partial w_{k_j}^k} + \ell^{-1} \sum_{k=1}^{\ell} \sum_{s=1}^{\ell} \sum_{k_i=1}^{n_k'} \sum_{s_j=1}^{n_s'} \frac{w_{k_i}^k w_{s_j}^s}{|w^k||w^s|} \frac{\partial \widetilde{F}}{\partial w_{k_i}^k} \frac{\partial}{\partial w_{s_j}^s}
\end{aligned}
\tag{76}
$$

In the last two terms, $|w^k| = |w^j|$, so it suffices to compute $|w^1|$.

Let $\binom{k_j}{k}$ denote the component of $c_k^{k_j}$ of $\nabla^{\mathcal{S}} \widetilde{F}_{\boldsymbol{w}}$. Label the first three terms on the last line of (76) by I, II, III. Then

1. In I, $\binom{k_j}{k} = \frac{\partial \widetilde{F}}{\partial w_{k_j}^k}$.

2. In II,

$$
\binom{k_j}{k} = \underbrace{-\frac{1}{|w^1|^2}}_{\text{independent of } k, k_j} \underbrace{\left( \sum_{k_i=1}^{n_k'} w_{k_i}^k \frac{\partial \widetilde{F}}{\partial w_{k_i}^k} \right) w_{k_j}^k}_{\text{independent of } k_j}.
$$

3. In III, by switching the $s$ and $k$ indices,

$$\binom{k_j}{k} = \underbrace{\frac{\ell^{-1}}{|w^1|^2} \left( \sum_{s=1}^{\ell} \sum_{s_i=1}^{n'_s} w^s_{s_i} \frac{\partial \widetilde{F}}{\partial w^s_{s_i}} \right)}_{\text{independent of } k, k_j} w^k_{k_j} \qquad \qquad \square$$

### I.2 Loss function gradient: Approach II

We show that in Approach II, formulas (54) and (76) are altered only by removing the $\ell^{-1}$ factor in the last term. For (54), we note that the metric $h$ in (49) is replaced by $\delta_{(k,k_i),(s,s_j)}$. Thus the proof of Prop. 5.2 changes to

$$\nabla^{Euc,k} \widetilde{F}_{w^k} = \frac{\partial \widetilde{F}}{\partial r_k} \frac{\partial}{\partial r_k} + \sum_{k_i=1}^{n_k-1} \frac{\partial \widetilde{F}}{\partial \psi_k^{k_i}} \frac{\partial}{\partial \psi_k^{k_i}},$$

and

$$\nabla^{\mathcal{S}} \widetilde{F}_{\boldsymbol{w}} = \left( \sum_{k=1}^{\ell} \frac{\partial \widetilde{F}}{\partial r_k} \right) \left( \sum_{s=1}^{\ell} \frac{\partial}{\partial r_s} \right) + \sum_{k=1}^{\ell} \sum_{k_i=1}^{n_k-1} \frac{\partial \widetilde{F}}{\partial \psi_k^{k_i}} \frac{\partial}{\partial \psi_k^{k_i}}$$

$$= \nabla^{Euc,\mathcal{W}} \widetilde{F} - \sum_{k=1}^{\ell} \frac{\partial \widetilde{F}}{\partial r_k} \frac{\partial}{\partial r_k} + \sum_{k=1}^{\ell} \sum_{s=1}^{\ell} \frac{\partial \widetilde{F}}{\partial r_k} \frac{\partial}{\partial r_s}$$

$$= \nabla^{Euc,\mathcal{W}} \widetilde{F} - \sum_{k=1}^{\ell} \sum_{k_i,k_j=1}^{n_k} \frac{w^k_{k_i} w^k_{k_j}}{|w^k|^2} \frac{\partial \widetilde{F}}{\partial w^k_{k_i}} \frac{\partial}{\partial w^k_{k_j}} + \sum_{k=1}^{\ell} \sum_{s=1}^{\ell} \sum_{k_i=1}^{n_k} \sum_{s_j=1}^{n_s} \frac{w^k_{k_i} w^s_{s_j}}{|w^k||w^s|} \frac{\partial \widetilde{F}}{\partial w^k_{k_i}} \frac{\partial}{\partial w^s_{s_j}}.$$

In the notation of the proof of Prop. 5.2, $\binom{k_j}{k}$ are unchanged for I and II, while for III

$$\binom{k_j}{k} = \frac{1}{|w^1|^2} \left( \sum_{s=1}^{\ell} \sum_{s_i=1}^{n'_s} w^s_{s_i} \frac{\partial \widetilde{F}}{\partial w^s_{s_i}} \right) w^k_{k_j}.$$

The same argument works for (76).

