# Generalized Tangent Kernel - Supplemental Materials

**Qinxun Bai**[*]                                    *qinxun.bai@horizon.auto*
*Horizon Robotics, Cupertino, CA, USA*

**Steven Rosenberg**[*]                              *sr@math.bu.edu*
*Department of Mathematics and Statistics, Boston University, Boston, MA, USA*

**Wei Xu**                                           *wei.xu@horizon.auto*
*Horizon Robotics, Cupertino, CA, USA*

**Reviewed on OpenReview:** *https://openreview.net/forum?id=HOnL5hjaIt*

## PyTorch Implementation of Approach I for ReLU MLP

The PyTorch implementation of the slice SGD for bias-free ReLU MLP under the GTK of Approach I is attached as follows, indeed, quite simple:

```python
import torch
import torch.nn as nn
from torch.optim.optimizer import Optimizer, required
import copy
import math

class SliceSGD(Optimizer):
    """Slice SGD Optimizer. """

    def __init__(self, params, lr=required,
                 momentum=0, weight_decay=0):
        defaults = dict(lr=lr, momentum=momentum, weight_decay=weight_decay)
        super().__init__(params, defaults)

        group = self.param_groups[0]
        params = group['params']
        self._depth = len(params)

        norm_prod = 1.
        for idx, param in enumerate(params):
            norm = param.data.norm()
            param.data.div_(norm)
            norm_prod *= norm

        norm_prod = norm_prod ** (1/self._depth)
        for param in params:
            param.data.mul_(norm_prod)

    def step(self):
        group = self.param_groups[0]
        params = group['params']
```

---

[*]Correspondence. These authors contributed equally to this work.

```python
pdata = [p.data / p.data.norm() for p in params]
layer_sum = [
    (data * p.grad.data).sum() for (data, p) in zip(pdata, params)]
layer_sum = torch.stack(layer_sum, dim=0)
coeff = (layer_sum.mean() - layer_sum)

norm_prod = 1.
for idx, param in enumerate(params):
    grad = param.grad.data + coeff[idx] * pdata[idx]
    param.data.add_(grad, alpha=-group['lr'])
    norm = param.data.norm()
    param.data.div_(norm)
    norm_prod *= norm

norm_prod = norm_prod ** (1/self._depth)
for param in params:
    param.data.mul_(norm_prod)
```