# OpenReview forum: "Generalized Tangent Kernel: A Unified Geometric Foundation for Natural Gradient and Standard Gradient"
_TMLR — Accepted by TMLR_

### Review · Reviewer_9Mtm · 2024-11-07

**Summary Of Contributions:**

This paper demonstrates that standard gradient descent captures the intrinsic geometric structure of the function space in function approximation, rather than the parameter space. The authors introduce a new mathematical object, the Generalized Tangent Kernel (GTK), and show that the GTK defines the Riemannian metric on the function space.

**Audience:**

Yes

**Broader Impact Concerns:**

No concerns.

**Claims And Evidence:**

Yes

**Requested Changes:**

None of my requested changes are crucial for securing my recommendation for acceptance, but I think they would help strengthen the work.

- As mentioned above, I think this paper could be of potentially broad interest, but it is not presented in a user-friendly way. I understand that the results are highly mathematical in nature, so this might be difficult. But I think interspersing simple, down-to-earth examples throughout the paper and clearly highlighting main results would help the reader along. This is not crucial for securing my recommendation for acceptance, but it would help strengthen the work.

- It could be interesting to connect your results with the very recent findings of [Shoji et al (2024)](https://arxiv.org/abs/2409.16422), who show that when an optimizer decreases a loss function, this implies that the iterates of the optimizer can be put into natural gradient form. Is it possible that a similar converse result holds for GTK, in the function space? This would be very interesting.

- It would be nice to connect this work to recent work on in-context learning (e.g., [Anh et al, 2023](https://arxiv.org/abs/2306.00297))--do your result apply to this setting as well? Or only supervised learning?

- "It is even more interesting to investigate the relation between choices of metrics and the generalization ability, which we have not touched in current work." This question has been studied in [Kozachkov et al (2023)](https://openreview.net/pdf?id=Sb6p5mcefw), with connections to NTK in section 4.3.

It would be great to see the above three papers (and related works) mentioned/discussed!

**Strengths And Weaknesses:**

- Main strength: The paper deals with fundamental mathematical aspects of (natural) gradient flows, and is therefore of broad potential interest.

- Main weakness: The paper is extremely dense, and is not easily accessible to someone  without deep working knowledge of Riemannian manifolds, RKHS, and Sobolev spaces. I found it difficult to follow for the most part.

---

> ### Author Response · Authors · 2024-12-02
> **Response to Reviewer 9Mtm**
>
> We are grateful to the reviewer both for the positive comments and for pointing out areas for improvement. We answer the bullet points in the Requested Changes section in order.
> - **Mathematical density of the paper.**
> This is addressed in our comments to all reviewers.  We are very sympathetic to this complaint, and have added simple examples in the text (see the end of Sec. 2.1, the end of Sec. 4.2, and the beginning of Sec. 5). A new paragraph is added in the Introduction to summarize/highlight the main theoretical results. We also added a pointer to the math background Appendices at the end of the Introduction.
> - **The Shoji *et al.* paper.**
> The Shoji *et al.* paper says that a first order ODE can be put into gradient flow form for some Riemannian metric, provided a loss function $L$ decreases along the flow of the ODE. This is a very interesting result and is now cited in the paper.
>
>     As a comment on the paper, the metric $h$ produced  is time dependent, so the flow equation $\dot\theta(t) = g(t, \theta(t))$ becomes $\dot\theta(t) = -\nabla^{h(t)} L(\theta(t)).$ The  more standard gradient flow equation $\dot\theta(t) = -\nabla^{h} L(\theta(t))$ has a time independent metric. The authors don't claim that every flow has such a decreasing function $L$. Indeed, the irrational flow on the torus and any flow in $\mathbb{R}^2$ with a limit cycle admit no such $L$.
>
>     If such an $L$ exists, it would be good to know if the ODE can be put into gradient flow form for a time independent metric.  It seems that this is possible on the regular set of $L$, as then the level sets $A_r = L^{-1}(r)$ (if nonempty) are codimension one submanifolds which are transverse to the flow lines of the vector field $\dot\theta$. Any Riemannian metric $h$ such that $T_\theta A_r \perp_h V$ and with $\Vert \dot\theta\Vert_h = -L'(\theta)$ has $\dot\theta = g(\theta,t) \Leftrightarrow \dot\theta(t) = -\nabla^{h} L(\theta(t)).$  Whether this construction can be extended to the singular set (which includes all the critical points for $L$) is unclear to us.
>
>     This allows us to tentatively answer the question of whether a similar result holds for GTKs.  In the language of the paper, we want to know: *given a loss function $L:\mathcal{M}\to\mathbb{R}$ and a vector field $V$ on $\Phi$ with $L$ decreasing along the flow lines of $V$,  is there a projection/GTK kernel $P:T_f\mathcal{M}\to T_{Pf}\Phi$ such that $V = P\nabla L$?* It seems to us that this question is equivalent to the question of putting the flow into (time independent) gradient form, since the existence of a GTK is equivalent to the choice of a Riemannian metric on the $\Phi$.  This remains to be worked out more carefully.
> - **The Anh *et al.* paper.**
> Transformers are also trained by standard (Euclidean) gradient descent, so in this sense, our work also applies to them. Regarding the in-context learning ability of transformers, whether there is a GTK explanation of the preconditioned gradient descent implemented by the forward pass of transformers is a very interesting question to study in future work. We added it to the discussion section Sec. 6.
> - **The Kozakchov *et al.* paper.**
> This paper establishes a very interesting relation between Riemannian contraction and an algorithmic stability generalization bound, so we also added it to the discussion section.  This paper also has a time dependent metric, so the comments above about the Shoji *et al.* paper are relevant.

---

> > ### Comment · Reviewer_9Mtm · 2024-12-02
> > **Thank you for reply**
> >
> > Thank you for your thorough reply, and for the edits to your paper. I believe they have improved the work.
> >
> > There is a very minor typo in the reference to the Shoji paper (page 15): the title is referenced incorrectly. It is showing up as " A note on reproducing kernels for Sobolev spaces". The arxiv link appears to be correct, though.

---

> ### Author Response · Authors · 2024-12-02
> **Thank you for the positive feedback and further comment**
>
> Thank you for catching this reference typo. We have updated the draft to fix typos in reference.

---

### Review · Reviewer_K4VW · 2024-11-15

**Summary Of Contributions:**

This paper proposes a formal study of natural gradient. First, it describes how to compute natural gradient using projection on the right tangent space. Then, it is shown that for classical settings in Machine Learning, natural gradient are actually not well defined (e.g. for empirical losses using maps on L2). Thus, it is proposed to study natural gradient on RKHS, where they can be well defined. This study motivates the introduction of a general kernel called the generalized Tangent kernel. Finally, the authors study the practical problem of optimizing a MLP with ReLU activation function, and address the problem of the non injectivity of the immersion using a sliced technique.

**Audience:**

Yes

**Claims And Evidence:**

Yes

**Requested Changes:**

While very interesting, some parts of the paper (such as Section 3) feel a bit like a draft with formal arguments for the computations.

In Section 5.1, it could be nice to add an Algorithm summarizing the procedure for the slice SGD (instead of the Pytorch code in Appendix, which should be in supplementary material).



Typos:
- Page 6: $\langle d_x, h^j\rangle_{\mathcal{H}_j}$

**Strengths And Weaknesses:**

**Strengths:**

This work answers interesting questions about Natural Gradients.

The paper is well organized. It introduces problems such as the ill defined gradients, and then provide solutions studying the problem on right spaces. It also provides an analysis of how to perform natural gradient descent for a concrete example (MLP with ReLU activation function), along with an experiment evidence of the fundness of the method.


**Weaknesses:**

The paper is very dense, and not always easy to follow. Also, there are lot of things in appendix, which could maybe help the reading (e.g. some backgrounds...).

A lots of arguments are only formal.

The experiments seem a bit preliminary and could be improved.

---

> ### Author Response · Authors · 2024-12-02
> **Response to Reviewer K4VW**
>
> We thank this reviewer for highlighting both strengths and weaknesses in the paper.  We have incorporated changes to all suggestions, which should improve the readability of the paper.  Here are specific responses to points raised in the Weaknesses and Requested Changes sections.
> - **Mathematical Density.**
> We have addressed this problem in the comments to all reviewers, hopefully to your satisfaction.
> - **Formal Arguments.**
> We provide formal computations in Sec. 3 to precisely pinpoint why functional gradients of empirical loss functions do not exist under common choices of metrics. The fact that delta functions appear in the formal gradient motivates the use of RKHS theory in Sec. 4 to make these computations rigorous.  We have added text in the fourth paragraph of the introduction and the first paragraph of Sec. 3 to highlight these points.
>
>     The calculations in other parts of this paper are rigorous, precisely because we use RKHS theory.
> - **Experiments.**
> This is discussed in the comments to all reviewers.
> - We have put the pytorch codes into Supplementary Material.
> - We corrected the typo on p. 6.

---

> > ### Comment · Reviewer_K4VW · 2024-12-02
> >
> > Thank you for the response and the revision. They have addressed my concerns.
> >
> > I think there is a small typo in Appendix B.1 when defining $df_p(X)$ (the $f$ is missing in the derivative with respect to $t$).

---

> > > ### Author Response · Authors · 2024-12-03
> > > **Thank you for your recognition and detailed proofreading**
> > >
> > > Thank you for catching this typo. We have updated the draft to correct it.

---

### Review · Reviewer_uCC2 · 2024-11-20

**Summary Of Contributions:**

The problem at hand is parametric approximation in spaces of functions using gradient flow to decrease some loss functional.
The authors introduce the generalized tangent kernel (GTK), which generalizes both the neural tangent kernel (NTK) and natural gradient. The GTK induces a Riemannian structure on the whole space of functions under consideration. It is shown that in many cases, natural gradients of empirical losses do not exist. Analysis of the case that the mapping of parameters to functions from a parametrized set is not injective is also presented, since this case appears frequently in practice.

**Audience:**

Yes

**Broader Impact Concerns:**

None.

**Claims And Evidence:**

Yes

**Requested Changes:**

Apart from the smaller issues I raise below, I think the most important requests are comparisons with similar ideas from the literature, writing the first part of the proof of the central Theorem 4.2 a bit clearer and addressing the last bullet point I listed under weaknesses.

# Small issues and typos
----

Abstract
---
-"under function approximation" seems strange to me, choosing a different preposition (e.g. for?) will certainly make this sentence easier to understand. The same goes for the first paragraph of the introduction.

Introduction
----
- In Figure 1, it should be $\mathscr{W}$ instead of $W$ to match the formatting of the caption.
- I would instead write "with the notable exception of van Oostrum".
- the term "functional gradient" is used without definition and I think (but I am not sure) that it is meant to mean the same as gradient.

Section 2
----
- It should be noted somewhere that the Einstein summation convention is used, since not everybody in this community will be familiar with it. However, only in equation (18) it is not used?
- In the last sentence before subsection 2.1 there is full stop missing at the end of the sentence.
- In the sentence before equation (2), I do not think that "Riemannian metric" is the correct terminology, since the manifold is infinite-dimensional.
- I do not understand the sentence "(6) is the pushforward $\phi_* g_E$ of the Euclidean metric", how can that formula be $\phi_* g_E$?

Section 3
----
- Even tough no confusion can arise, I would prefer if the summation indices would not be (partially) omitted in formulas (9) - (13), (16).
- In the new paragraph after equation (8) it should be " a curve $(f_t)_{t \ge 0} \subset \mathcal{M}$" instead.
- In the paragraph after remark 3.1. you first write "metric on $\Phi$" and then "metric on $T_p \mathcal{P}$", which is inconsistent.
- I would remove the word "easily" from the last sentence in this section, because it is redundant.
- In the example on the bottom of page 5 about the Fisher-Rao metric assumptions on $M$ are missing. If $M$ is not a compact Riemannian manifold in $\mathbb R^d$ without boundary, then e.g. the identification of the tangent space is not true, see section 3 of Lafferty's 1988 paper "The Density Manifold and Configuration Space Quantization".

Section 5
----
- In 3. at the beginning of the section, "Corollary" is misspelled.

Section 6
-----
- "Euclidean" is misspelled.


Bibliography
----
- Names of conferences (like NeurIPS) and Journals (like Information Geometry) should be capitalized.
- The book title by Schmüdgen is missing an 's' at the end.

Appendix B.1
----
- When defining functions, I recommend using \colon for so that the spacing is correct, e.g. $f \colon M \to N$ instead of $f : M \to N$.
- In my opinion, sentences should not be started with a mathematical symbol (this is not the only place where this occurs, e.g. after equation (1)), which is why I would advise to start the third paragraph by "A (Hausdorff second countable) space $M$ is a Riemmanian...".
- "a slight perturbation of the standard dot product" is not very precise, I do not really understand how to understand this.
- After "It follows that any tangent vector" it should by $T_M$, not $T_m$.

Appendix B.2
-----
- Two formulas before equation (28), the last differential should be $\xi^n$, not $x^n$.
- Before equation (29), the term "isomorphic" is misspelled and $\text{d}\;\text{vol}$ is only used there, which is not consistent with the rest of the section and paper (compare e.g. to equation (8)), especially since the Sobolev spaces that are considered two sentences earlier are on $\mathbb R^n$, not $\Omega$.

Appendix B.3
----
- I recommend getting rid of "except on a countable set", since there are sets of Lebesgue measure zero that are uncountably infinite an more generally, one does not have to choose the Lebesgue measure anyway.

Appendix C
----
- I think the terminology "gradient flow" for the a curve $\gamma \colon (0, \infty) \to X$ is more common than "gradient flow line".

Appendix I
----
- I suppose it should be "function space" instead of "functional space" in the first sentence.

**Strengths And Weaknesses:**

# Strengths
- The paper is fairly well-written and clear and the idea of GTK seems novel.
- An effort is made to include all the necessary background in the appendix (which I however, due to time constraints, could not read in its entirety)

# Weaknesses

- More discussion of the literature is would be helpful, for example explaining differences to the ideas of Bai et al.'s "A Geometric Understanding of Natural Gradient" (arXiv:2202.06232) or Bastian et al's "Sobolev gradients and neural networks".
- The extra computation cost of $O(n)$ seems not insignificant, and it would be good if the authors could comment a bit more on this issue.
- I appreciate that numerical examples were conducted, however the advantage of the proposed method as shown in Figure 2(a) seems negligible. The paper would benefit from more numerical experiments or different ones, where differences become more apparent or a discussion on why they are not.
- I do not understand why in the very beginning of section 2 you equip $\mathcal{M}$ with a Sobolev or Fréchet topology (which seems a reasonable choice to me), but then in section 3 choose the less natural $L^2$-topology to show non-existence of natural gradients. In other words, if one would instead choose the Sobolev of Fréchet topology or consider these functions as distributions, do natural gradients of empirical losses exist and thus no RKHS approach is necessary? I think at least for the topology induced by viewing the functions as a subset of (tempered) distributions, gradients should exist by the very argument given in the paper.

# Questions
- In theorem 4.2. should $X$ be instead be $M$? If not, what is $X$, is it a finite set?
- In Lemma 2.1 (i), what is the "*flow* of $P(\nabla F)$"? Do you mean a gradient flow?
- In section 2.2. what is meant by "flow lines for $\nabla F$ *do not project* to flow lines of $S \circ \nabla F$"? Are you claiming that there is no projection mapping one to the other?
- In Proposition 4.1, why is $T_f \Phi$ a RKHS?

---

> ### Author Response · Authors · 2024-12-02
> **(1/2) Response to Reviewer uCC2**
>
> We are extremely grateful to this reviewer for the remarkably detailed reading of the paper and feedback. We have gone through every comment and addressed all of them.
>
> ## Weaknesses
> We answer the reviewers bullet points in order.
> - **Comparison to Bai *et al.* and Bastian *et al.***
> Bai *et al.* is an unpublished preprint. In addition, the current submission goes beyond Bai *et al.* by formally introducing the concept of GTK, addressing the non-immersion case of natural gradients, and conducting further experiments. The paper of Bastian *et al.* is doing either Euclidean or natural gradient flow in a function space which keeps track of the first $k$ derivatives of a function. However, the Sobolev $k$ norm is not explicitly used, so this paper does not do gradient flow on a Sobolev space w.r.t. the Sobolev norm.  So we would describe this paper, while quite interesting, as doing ``Sobolev-inspired'' gradient flow and not directly related to our paper.
> - **Extra computation cost.**
> While the extra $O(n)$ computational cost seems non-negligible, as shown in Prop. 5.2 and the pytorch codes (now in supplementary materials based on Reviewer's suggestion), these extra computations are simple matrix multiplications/additions between network weights and their standard gradients (already) obtained by backpropagation. This parallel nature can be effectively leveraged by modern GPU's, and is fundamentally different from the sequential nature of backpropagation. Therefore, as shown in Figure 2, the extra cost over standard gradient descent is negligible in practice. We have added a short discussion of this in Sec. 5.2.
> - **Experimental results.**
> This has been addressed in the comments to all reviewers.
> - **Choices of topology and reasons for working with RKHS.**
> The purpose of Sec. 3 is to precisely show the nonexistence of natural gradients for common choices of metrics in ML, such as $L^2$ and Fisher-Rao. We opt for the Sobolev or Fr&eacute;chet topology in Sec. 2 to make our theoretical setup mathematically rigorous. While these choices (and  others) solve the nonexistence problem, using RKHS theory is appealing for two reasons:
>   1. As mentioned in Remark 3.1 (and emphasized more  in the revised text), whenever a loss function measures pointwise information, the gradient will formally contain delta functions. To handle these gradients, delta functions must be continuous linear functionals on the function space, which is precisely the definition of an RKHS.  In other words,    the natural gradient   for pointwise loss functions on a Hilbert space/manifold exists iff the tangent spaces are RKHSs.
>   2. As a result, the RKHS perspective unifies all projected functional gradients, including both standard gradients and natural gradients. This leads to the key concept of this work, GTK and its automatic orthonormality property, which is based on  RKHS theory.
>
>     In contrast, while being theoretically sound, the practical computation of natural gradients under Sobolev topologies is currently intractable, and even more difficult for the Fr&eacute;chet topology coming  from a countable family of seminorms. As shown in Appendix G, even after computing the exact reproducing kernel of Sobolev spaces (Lemma G.1), we have to rely on a series of approximations to actually implement the Sobolev natural gradient. Note that one step in our approximation scheme also depends on RKHS theory.
>
>     These remarks apply to the space of tempered distributions (another Fr&eacute;chet space/RKHS).  Computing the gradient there is an interesting theoretical question, but implementing this in practice will probably be difficult.
> ## Other Requested Changes
> - **A clearer proof of Theorem 4.2.**
> Thank you for bringing this up. We have updated the proof of the central Theorem 4.2 to make it clearer.
> ## Questions
> - In Theorem 4.2, $X$ is any set.  We have added this to the theorem statement.
> - In Lemma 2.1(i), $P(\nabla F)$ is a vector field on $\Phi$, and so determines a flow on $\Phi.$ The content of this part of the Lemma is that this flow is indeed the gradient flow for $F|_{\Phi}.$  So the statement that the flow of $P(\nabla F)$ is a gradient flow takes a short proof.
> - Thanks for catching this poorly worded sentence.  We have replaced "Thus flow lines for $\nabla F$ do not project to flow lines of $S\nabla F.$" with "Thus Lemma 2.1(i) fails, and the flow lines for $\nabla(F|_\Phi)$ are not related to the flow lines of $S\nabla F.$"
> - Thanks for this question. In Proposition 4.1, $T_f\Phi$ is a finite dimensional subspace of $T_f\mathcal{M}(M, \mathbb{R}^n) \simeq \mathcal{M}(M,\mathbb{R}^n)$.  Since $\mathcal{M}$ is an RKHS, so is $T_f\Phi$. We added this as a comment after the Proposition.

---

> ### Author Response · Authors · 2024-12-02
> **(2/2) Response to Reviewer uCC2**
>
> ## Small issues and typos
> ### *Abstract*
>   - This has been changed.
> ### *Introduction*
>   - Figure 1 has been updated to match the formatting of the caption.
>   - in "van Oostrum" has been changed to "of van Ostrum."
>   - The term "functional gradient" appears in ML literature, and just means the gradient of a loss function on a function space.  To avoid confusion, we changed the language in the Abstract and Introduction, and added a footnote on p. 2 to define functional gradient.
> ### *Section 2*
>   - We added a sentence about Einstein summation convention before Proposition 2.1.  Usually, this convention applies when an index in a tensor appears once as a superscript and once as a subscript, as in (17).  In (4), both indices are subscripts, so we include the summation sign.  (We could be purists and rewrite the right hand side of (4) as
> $\sum_{i,j}\delta^{ij}\langle\nabla F, h_i\rangle h_j$,
>  but this seems more confusing to us.) As a result, we've added  summation signs in (4) and (25),
>  and kept  the summation in (18).  However, in (13) - (15), we want to keep $\sum_i$, since this is not a sum over tensor indices. We understand this may not be strictly consistent, but we are trying to balance strict consistency  with readability. Similarly, in the first comment for Sec. 3, you recommend adding summation signs everywhere;  we prefer not to, but of course will do so if you feel strongly about this.
>   - Full stop added at the end of Sec. 2.1.
>   - We believe the terminology "Riemannian metric" is often used for smooth infinite dimensional manifolds $\mathcal M$. It just means that each tangent space has a positive definite inner product, *i.e.,* each tangent space is a Hilbert space, and that the inner product varies smoothly.  To define "varies smoothly", since $\mathcal M$ is locally diffeomorphic to an open ball in a Hilbert (or Banach or Fr&eacute;chet) space, the tangent spaces are locally isomorphic to a fixed infinite dimensional  vector space.  Then the smoothness of the inner products is defined in terms of their Fr&eacute;chet derivatives. For Hilbert and Banach spaces, this is treated carefully in Lang's *Introduction to Differential Topology* and Klingenberg's *Riemannian Geometry*. For Fr&eacute;chet spaces, the standard reference is Kriegl-Michor's *The Convenient Setting in Global Analysis*.
>   - We changed the last sentence in Sec. 2.2 to "Since ..., the Riemannian metric determined by (6) is the pushforward $\phi_*g_E$ of the Euclidean metric on $\mathcal W$." We hope this is ok. The point is that for $v = v^i (\partial \phi/\partial w^i), z = z^j (\partial \phi/\partial w^i),$ (6) determines that $\langle v,z\rangle_{\phi_*g_E} = \sum_i v^iz^i.$
> ### *Section 3*
>   - This is addressed in comment 1. for Sec. 2.
>   - Thanks for this correction.
>   - Thanks again.  We changed "The Fisher-Rao metric on  $\Phi$ ...." to "The Fisher-Rao metric at  $T_{\phi(w)}\Phi$ ...."
>   - "easily" has been removed.
>   - Thanks for pointing this out.  We cleaned up the language and added the 1988 Lafferty paper to the references.
> ### *Sections 5 and 6.*
>   - The typos are corrected.
> ### *Bibliography*
>   - These two comments have been addressed.
> ### *Appendix B.1*
>   - We have changed : to $\backslash$colon in the context of $f\colon X\to Y$ throughout the paper.
>   - We have changed the sentence starting with a math symbol in Appendix B.1, and we checked the rest of the paper for other such sentences.
>   - We clarified the sentence about perturbing the standard dot product.
>   - We believe our notation is correct. We did change $r_0$ to $n$ in the second $T_m f^{-1}(n).$
> ### *Appendix B.2*
>   - Thanks for pointing out the typos and the poor notation.
> ### *Appendix B.3*
>   - The first two sentences have been reworded.
> ### *Appendix C*
>   - To our thinking, a (negative) gradient flow line is a curve $\gamma(t)$  soling $\dot\gamma(t) = -\nabla F_{\gamma(t)},$  while a gradient flow is the collection of all such curves.  For example, Lemma 2.1(i) discusses the gradient flow,  while Lemma 2.1(ii) discusses a gradient flow line. So we prefer to leave the language in Appendix C as is, but this is not a line in the sand.
> ### *Appendix I*
>   - Thank you for catching this. We have updated "functional space" to "function space" throughout the paper.

---

### Author Response · Authors · 2024-12-02
**Overall Response**

We thank all reviewers for their constructive suggestions and comments. All requested changes have been addressed and incorporated into the updated draft in **blue** (except for one word changes). Some common questions are addressed as follows.

**Dense presentation.** We agree with the reviews that our paper ends up quite dense due to the nature of the content. We appreciate the reviewers'  helpful suggestions to make the paper more  accessible to the ML community, and have incorporated all of them. In particular, we point the readers at the end of the Introduction to Appendices A and B for background material in Riemannian geometry, Sobolev spaces, and RKHS theory. We inserted simple examples in Sec. 2.1, Sec. 4.2, and the beginning of Sec. 5. We also added an algorithm table in Sec. 5.1 summarizing the slice SGD procedure.

**Preliminary experiments.** We agree with the reviews that our experimental results are preliminary. We would like to emphasize a few points regarding the role of experiments in this work.
1. Our main focus is studying several long-overlooked theoretical aspects of functional gradients and their projections onto the parameterized embedding space. The major outcome of this study is the introduction of a mathematical object (GTK) with appealing geometric properties that define new Riemannian structures on the whole function space, unifying standard gradient, natural gradient, and beyond.
2. Besides theoretical contributions, our theory also leads to new families of natural gradient variants. We implemented and tested a few exemplars  to show the potential and practicality of our theoretical results. We feel that exploring the full potential of this new theoretical framework belongs to future work.
3. In our preliminary experiments, while the improvement of our methods over baselines seems marginal, it is worth noting that our experimental settings and hyper-parameters follow the best practice of baseline methods, which have been extensively optimized over the years by the ML community. Our methods simply align everything with baselines without specific tuning. We feel that these first results show the potential of our theory, while leaving a large room for practical optimization to future work. We have added discussion on this point in the main text.

---

### Decision · Action_Editor_ksp6 · 2025-01-26

**Recommendation:** Accept as is

**Comment:**

The paper has been well-received by all the reviewers, and the revised version has successfully addressed their concerns. However, a minor issue remains regarding the lack of practical applications or examples in the current version. Please include all the promised changes.

**Audience:**

Yes, appropriate for the audience.

**Claims And Evidence:**

The paper's claims are supported by evidence, with its primary assertion focusing on the generalization of the Neural Tangent Kernel (NTK). It provides the necessary framework to achieve this goal.